# *Urocortin-3* neurons in the mouse perifornical area promote infant-directed neglect and aggression

Anita E Autry[1,2], Zheng Wu[1†], Vikrant Kapoor[1], Johannes Kohl[1‡], Dhananjay Bambah-Mukku[1], Nimrod D Rubinstein[1§], Brenda Marin-Rodriguez[1], Ilaria Carta[2], Victoria Sedwick[2], Ming Tang[1,3], Catherine Dulac[1]*

[1]Howard Hughes Medical Institute, Department of Molecular and Cellular Biology, Center for Brain Science, Harvard University, Cambridge, United States; [2]Dominick P. Purpura Department of Neuroscience, Department of Psychiatry and Behavioral Sciences, Albert Einstein College of Medicine, Bronx, United States; [3]FAS Informatics Group, Harvard University, Cambridge, United States

*For correspondence:
dulac@fas.harvard.edu

Present address: [†]Mortimer B. Zuckerman Mind Brain Behavior Institute and Department of Neuroscience, Columbia University, New York, United States; [‡]The Francis Crick Institute, London, United Kingdom; [§]Calico Life Sciences, South San Francisco, United States

**Abstract** While recent studies have uncovered dedicated neural pathways mediating the positive control of parenting, the regulation of infant-directed aggression and how it relates to adult-adult aggression is poorly understood. Here we show that *urocortin-3* (*Ucn3*)-expressing neurons in the hypothalamic perifornical area (PeFA[Ucn3]) are activated during infant-directed attacks in males and females, but not other behaviors. Functional manipulations of PeFA[Ucn3] neurons demonstrate the role of this population in the negative control of parenting in both sexes. PeFA[Ucn3] neurons receive input from areas associated with vomeronasal sensing, stress, and parenting, and send projections to hypothalamic and limbic areas. Optogenetic activation of PeFA[Ucn3] axon terminals in these regions triggers various aspects of infant-directed agonistic responses, such as neglect, repulsion, and aggression. Thus, PeFA[Ucn3] neurons emerge as a dedicated circuit component controlling infant-directed neglect and aggression, providing a new framework to understand the positive and negative regulation of parenting in health and disease.

## Introduction

Mammals invest considerable resources to protect and nurture young offspring. However, under certain physiological and environmental conditions, animals neglect or attack young conspecifics. Specifically, males in some species kill unfamiliar infants to gain reproductive advantage (*Hrdy, 1974*; *Hrdy, 1979*; *Parmigiani and Vom Saal, 1994*), and females in many species are known to cull or neglect their young during stressful circumstances such as food shortage or high risk of predation (*Svare and Bartke, 1978*; *Heasley, 1983*; *Marsteller and Lynch, 1987*; *König, 1989*; *Lukas and Huchard, 2018*). Notably, stress in humans of both sexes has been identified as a major risk factor for peripartum mental disorders and associated impairments in parent-infant interactions (*Bloch et al., 2003*; *Scarff, 2019*; *Hutchens and Kearney, 2020*).

In species with biparental care, male's onset of parenting behavior and loss of infant mediated aggression typically coincide with the timing of birth of offspring following mating (*vom Saal, 1985*; *Tachikawa et al., 2013*; *Wu et al., 2014*). This phenomenon of time-dependent parental care is observed in biparental mammals, as well as more widely throughout the animal kingdom, such as insects, fish, and birds (*Elwood, 1994*).

Recent studies have investigated the neural mechanisms underlying parental behavior and uncovered specific circuit nodes and neuronal populations mediating the positive control of parenting (*Wu*

*et al., 2014*; *Marlin et al., 2015*; *Scott et al., 2015*; *Fang et al., 2018*; *Kohl et al., 2018*; *Stagkourakis et al., 2020*). However, the existence of a similar circuit organization to control infant-directed aggression and the relationship between such pathway and that of adult-mediated aggression have not yet been determined (*Tachikawa et al., 2013*; *Tsuneoka et al., 2015*; *Amano et al., 2017*; *Chen et al., 2019*; *Sato et al., 2020*). Using induction of the immediate early gene (IEG) *Fos* as a molecular readout of neuronal activation after interaction with pups, we found that *urocortin-3* (*Ucn3*)-expressing neurons in the perifornical area (PeFA^Ucn3^) of the hypothalamus are activated during infant-directed attacks in male and female mice, but not during other forms of aggression. Inhibition of PeFA^Ucn3^ neurons in virgin males suppresses infant-directed aggression, while activation of these cells in virgin females disrupts parental behavior. We determined the input and output organization of PeFA^Ucn3^ neurons and showed that optogenetic activation of PeFA^Ucn3^ axon terminals in specific target regions, that is, ventromedial hypothalamus (VMH), ventral lateral septum (LSv), and amygdalohippocampal area (AHi), triggers different aspects of infant-directed agonistic responses, such as neglect, repulsion, and aggression. Based on their function and connectivity, PeFA^Ucn3^ neurons help define a new brain pathway mediating the specific expression of infant-directed neglect and aggression. These findings provide new mechanistic insights into the control of opposite behaviors toward infants according to an animal's sex and physiological status.

## Results

### *Urocortin-3*-expressing cells in the perifornical area are specifically activated during infant-directed aggression

To identify neuronal populations involved in infant-directed aggression, we monitored the induction of the IEG *Fos* across brain regions involved in social behavior control, that is, the hypothalamus and septal and amygdaloid nuclei, in infanticidal virgin males, fathers and mothers after interactions with pups (*Figure 1A and B*). Significantly more *Fos*+ cells were identified in the medial bed nucleus of stria terminalis (BNSTm), the LSv, and perifornical area (PeFA) of infanticidal males compared to mothers and fathers, with the most robust differences in *Fos* expression between infanticidal and parental animals observed in the PeFA, a small nucleus between the fornix and the paraventricular hypothalamus (PVH; *Figure 1A and B*). By contrast, only rare *Fos*+ cells were observed in the medial preoptic area (MPOA) of infanticidal males compared to mothers, as expected from previous studies (*Wu et al., 2014*; *Kohl et al., 2017*; *Figure 1A and B*).

Based on the abundant literature highlighting the critical role of hypothalamic nuclei in infant-adult interactions, we focused our study on the role of the PeFA in infanticidal behavior and performed an expression analysis to characterize the identity of PeFA cells activated by infant-directed attacks. For this purpose, we visualized *Fos* expression by in situ hybridization in tissue sections from virgin males after interactions with pups and performed laser-capture microdissections of the corresponding *Fos*+ areas in adjacent sections left untreated, followed by gene expression analysis of this material (*Figure 1C*).

Expression analysis revealed 140 transcripts with significant differential expression between the *Fos*+ area and the adjacent dorsolateral *Fos*-negative hypothalamic tissue. Among them, *vasopressin* (*Avp*) and *urocortin-3* (*Ucn3*) were the most upregulated transcripts, followed by *oxytocin* (*Oxt*), *thyrotropin releasing hormone* (*Trh*), and *galanin* (*Gal*), each with a more than twofold change (log2) in gene expression relative to the dorsolateral *Fos*-negative region (*Figure 1D*; p<0.001, adjusted p-value<0.05). Phosphorylated-S6 ribosomal pulldown (*Knight et al., 2012*) after pup-directed attack further confirmed the enrichment in *Ucn3* expression in neurons activated by infanticide (*Figure 1—figure supplement 1A*). In situ hybridization analysis showed that *Ucn3* expression is spatially restricted to the PeFA and is not expressed in the paraventricular nucleus nor in other surrounding brain areas (*Figure 1E*). This is in agreement with previous studies describing specific expression of *Ucn3* in the PeFA, as well as in few other brain regions such as the posterior bed nucleus of the stria terminalis (pBNST), the MPOA, the medial amygdala, and the nucleus ambiguus (*Venihaki et al., 2004*; *Wittmann et al., 2009*; *Deussing et al., 2010*). Overall, *Ucn3* displayed the highest colocalization with *Fos*+ cells after infanticide (55.3% *Fos*+ cells co-express *Ucn3*) compared to *Avp* (7.5% *Fos*+ cells co-express *Avp*; *Figure 1—figure supplement 1B*). Significant expression overlap was also seen between *Fos* and *Trh* (*Figure 1—figure supplement 1B*), consistent with a previous report

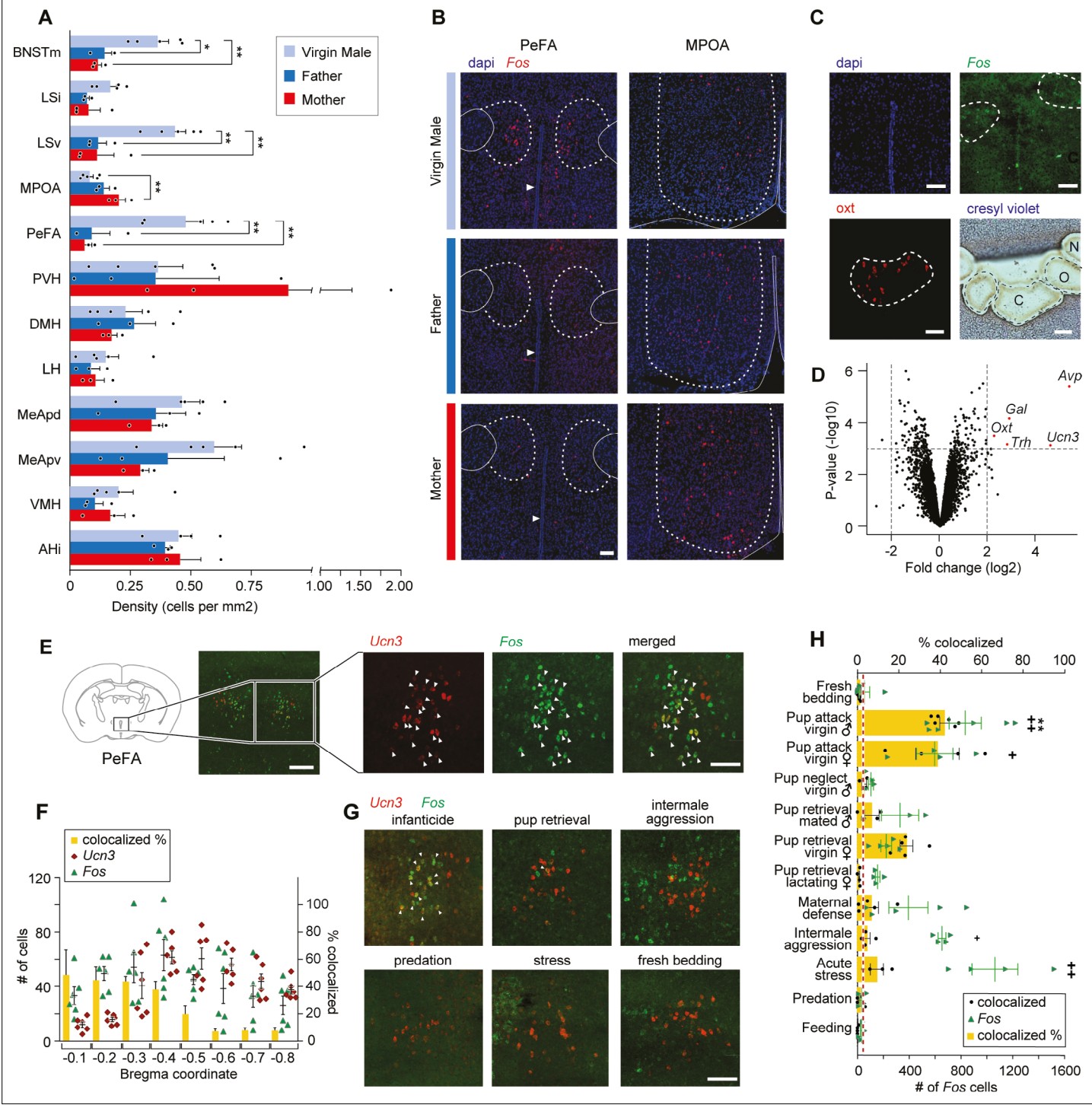

**Figure 1.** The neuropeptide gene *urocortin-3* (*Ucn3*) in the perifornical area (PeFA) marks neurons activated during infanticide. (**A**) Quantification of *Fos+* cells per mm$^2$ in brain areas of interest (oriented from top to bottom, rostral to caudal) in virgin males (attacking), mated males (parental), and mated females (parental) after pup exposure. One-way ANOVA followed by Tukey's multiple comparison test reveals significant differences in the number of *Fos+* cells identified between parental and infanticidal animals in medial bed nucleus of stria terminalis (BNSTm) ($F_{2,8} = 12.47$, $p<0.0035$), ventral lateral septum (LSv) ($F_{2,8} = 14.31$, $p<0.0023$), medial preoptic area (MPOA) ($F_{2,8} = 8.536$, $p<0.0104$), and PeFA ($F_{2,8} = 12.02$, $p<0.0039$) (virgin male n = 5; fathers n = 3; mothers n = 3). (**B**) Representative in situ hybridization images of *Fos* (red channel) expression in PeFA and MPOA after animal exposure to pups with counterstain DAPI (blue channel) (scale bar: 100 µm). Dotted lines in right and left denote the boundary of brain regions of interest (PeFA and MPOA, respectively), solid lines denote the fornix, arrowheads indicate location of third ventricle in the left panels, and solid lines denote the edge of the brain and third ventricle in the right panel. (**C**) Laser-capture microscopy strategy to identify markers of cells activated during

*Figure 1 continued on next page*

*Figure 1 continued*

infanticide (scale bar 100 μm). Tissue material corresponding to areas labeled by *Fos* (C) after pup-mediated attack, as well as oxytocin+ (Oxt) (O) and adjacent negative (N) area was laser-dissected and the corresponding transcripts characterized by microarray analysis. (D) Volcano plot showing the fold change (log base 2) against the p-value (log base 10) for all genes. Transcripts with greater than a twofold change in expression in the *Fos*+ compared to negative region and a p-value less than 0.001 are shown in red: *vasopressin* (*Avp*), *thyrotropin-releasing hormone* (*Trh*), *galanin* (*Gal*), *Oxt*, and *Ucn3* (n = 3 biological replicates, each containing pooled tissue from six males). (E) Representative in situ hybridization showing colocalization of *Ucn3* (red) and *Fos* (green) expression after infanticide (scale bar 100 μm), arrowheads indicate colocalization. (F) Plot depicting number (#) of neurons expressing *Fos* (green triangles), *Ucn3* (red circles) and percent of cells expressing both *Fos* and *Ucn3* (yellow bars) per brain section across Bregma coordinates (n = 6) indicate high colocalization in rostral PeFA. Overall representation of *Ucn3*+ neurons in the PeFA (2.74%) is indicated by the red line. (G) Representative in situ hybridization showing expression of *Ucn3* (red) and *Fos* (green) after infanticide and other behaviors in the rostral PeFA, with colocalization (white arrowheads) only observed after infanticide (scale bar 100 μm). (H) Percentage of *Ucn3*+ cells colocalized with *Fos* after various behaviors (yellow bars; *significance) and number of *Fos* cells induced by various behaviors across sections of the rostral PeFA (green triangles; significance #) (n = 3–6/ group). Kruskal–Wallis test followed by Dunn's multiple comparisons test reveals significant differences in number of *Fos*+ cells in rostral PeFA between fresh bedding control exposure and pup attack in virgin males and virgin females, intermale aggression, and acute stress (#p<0.0001); yet, despite the observed widespread activation of PeFA cells during various behaviors, only pup-directed attacks by virgin males induced significant *Fos* expression in PeFA$^{Ucn3}$ neurons (*p<0.0002).

The online version of this article includes the following figure supplement(s) for figure 1:

**Figure supplement 1.** Expression of halorhodopsin in PeFA urocortin-3 neurons.

of co-expression of *Ucn3* and *Trh* by large fractions of PeFA neurons (*Wittmann et al., 2009*). By contrast, few neurons active during infanticide were marked by *Gal* (20.8%), a marker that is highly co-expressed with *Fos* (42.8%), as detected by our microarray (*Figure 1D*) and pS6 data (*Figure 1— figure supplement 1A*, B), but that labels only a very small subset of PeFA neurons. Another candidate marker, *Oxt*, did not colocalize significantly with *Fos* in mice that committed infanticide (*Figure 1— figure supplement 1B*), and may have been mistakenly identified due to its high level of expression in this region of the hypothalamus. Quantification of *Ucn3* and *Fos* colocalization across the entire PeFA revealed that while *Ucn3*+ cells are found throughout the PeFA, *Ucn3*+/*Fos*+ neurons identified after infant-directed attack are located primarily in the rostral half of the PeFA (*Figure 1E and F*).

Next, we asked whether the activity of *Ucn3*+ PeFA neurons (PeFA$^{Ucn3}$) neurons was specifically associated with infant-directed agonistic interactions or was more generally increased by aggressive displays. We found that a significant number of PeFA neurons were activated by pup-directed attack in virgin males and females as well as by acute stress (*Figure 1H*; green dots). However, remarkably, when looking specifically at the *Ucn3*+ subpopulation in PeFA, *Fos* expression was only significantly induced in PeFA$^{Ucn3}$ neurons after pup-directed attacks by virgin males compared to fresh bedding exposure. A trend was also observed in infanticidal virgin females, resulting from the analysis of three rare infanticidal mice seen out of a large group of otherwise non-aggressive virgin females. Strikingly, no *Fos* induction was observed in PeFA$^{Ucn3}$ neurons after other forms of aggression such as maternal defense, inter-male aggression, and predation, nor by stress or feeding, suggesting a specific role of this population in infant-directed aggression (*Figure 1G and H*; yellow bars). In addition, while *Ucn3*+ cells did not express *Fos* in lactating females after exposure to pups, a non-statistically significant trend toward activation of *Ucn3*+ neurons in virgin females that either attacked or retrieved pups was observed, suggesting some engagement of this population in virgin states (*Figure 1H, Figure 1—figure supplement 1E*). No sex or state-dependent difference was observed in the number of PeFA$^{Ucn3}$ neurons in virgin or mated males and females (*Figure 1—figure supplement 1C*), and most PeFA$^{Ucn3}$ neurons expressed the vesicular glutamate transporter 2 (*Vglut2*, or *Slc17a6*; 91.3% ± 1.2%) suggesting an excitatory function (*Figure 1—figure supplement 1D*).

We conclude from these data that pup-directed aggression, but no other forms of adult aggression or other behaviors, specifically activates excitatory *Ucn3*+ neurons of the rostral PeFA.

## Inhibition of PeFA$^{Ucn3}$ neuronal activity in virgin males blocks infanticide

To test if PeFA$^{Ucn3}$ neuronal activity is required for the display of male infanticide, we used a conditional viral strategy to express the inhibitory opsin NpHR3.0, or yellow fluorescent protein (YFP), in PeFA$^{Ucn3}$ neurons and implanted bilateral optical fibers above the PeFA of virgin males (*Figure 2A, Figure 2—figure supplement 1*). Before laser stimulation, control and NpHR3.0-injected males showed similarly high levels of pup-directed aggression with short latencies to attack (*Figure 2B–D*). By contrast, optogenetic inhibition of PeFA$^{Ucn3}$ neurons led to a reduction in the number of males

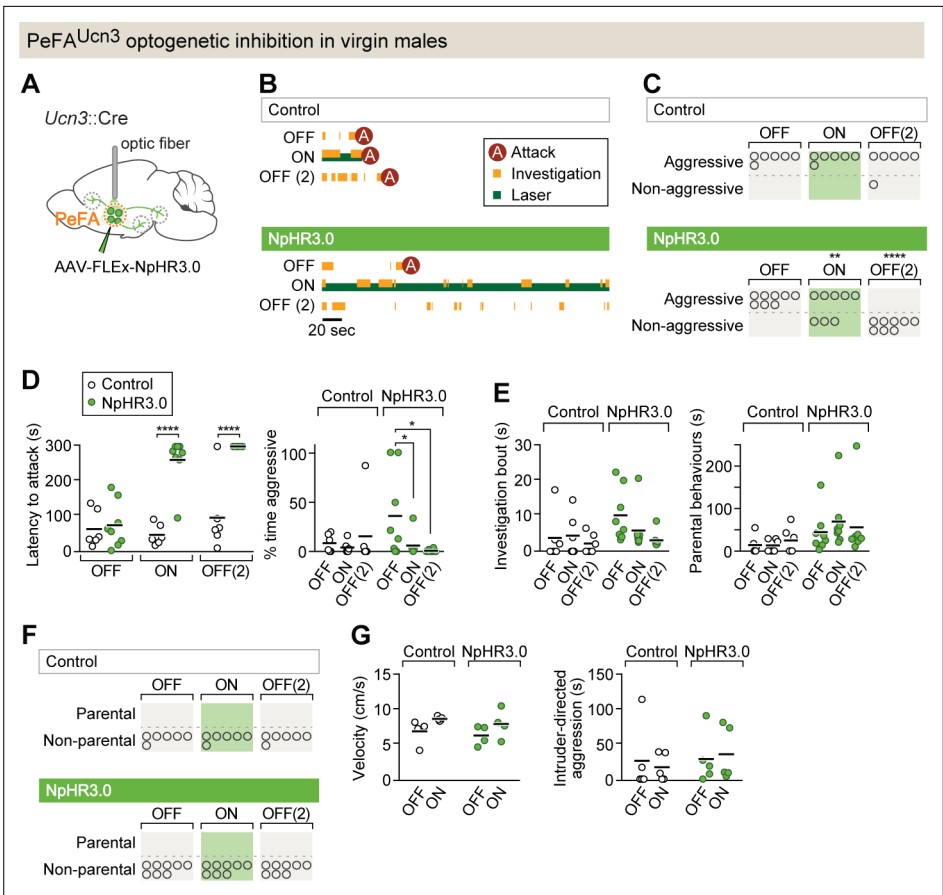

**Figure 2.** PeFA$^{Ucn3}$ neurons are required for infant-directed aggression in virgin males. (**A**) Setup for optogenetic silencing of PeFA$^{Ucn3}$ neurons in infanticidal virgin males. (**B**) Representative behavior trace of a control and a NpHR3.0-expressing male across three consecutive sessions (scale bar 20 s of 5 min test). (**C**) Numbers of control and NpHR3.0-expressing males showing aggressive pup-directed behavior (accelerated failure regression model with first session as time-point baseline and controls as treatment baseline, significant difference between control (n = 6) and NpHR3.0 (n = 8) during laser inhibition p<0.01 and during second laser OFF session p<0.0005). (**D**, left) Pup-directed attack latency across sessions comparing control (open dots) and NpHR3.0 (green dots)-infected males. Two-way repeated measures ANOVA control vs. NpHR3.0 $F_{(1,12)}$ = 35.05, p<0.0001, sessions $F_{(2,24)}$ = 22.71, p<0.0001, and interaction $F_{(2,24)}$ = 17.30, p<0.0001 followed by Bonferroni's multiple comparisons test. (Right) Percentage of time displaying aggression toward pups across three sessions in control (open dots) and NpHR3.0 (green dots) males. Two-way repeated measures ANOVA showed no main effect while post-hoc Tukey's multiple comparisons test revealed significant differences between sessions in the NpHR3.0 group. (**E**, left) Pup investigation bout length across three sessions in control and NpHR3.0 males. (Right) Duration of parental care behaviors across three sessions in control and NpHR3.0 males. (**F**) Number of males in control and NpHR3.0-expressing groups displaying parental behaviors. (**G**, left) Locomotion velocity in control and NpHR3.0-expressing males with and without laser stimulation. (Right) Duration of aggressive displays during adult male interactions in control and NpHR3.0-expressing males with and without laser stimulation.

The online version of this article includes the following figure supplement(s) for figure 2:

**Figure supplement 1.** Representative image of AAV-mediated NpHR3.0 expression in the periforniral area (PeFA) of *Ucn3*::Cre males (scale bar 100 µm).

displaying pup-directed attacks, an increase in attack latency, and a decrease in the duration of aggressive displays toward pups (*Figure 2B–D*). Strikingly, the suppression of pup-directed aggression persisted – and even further increased – within the next 24 hr till at least the second 'laser OFF' session, suggesting a prolonged effect of acute PeFA$^{Ucn3}$ neuronal silencing ('OFF(2)' in *Figure 2B–D*). Males spent equivalent time investigating pups in all conditions and did not increase their parental behaviors within the test period (*Figure 2E and F*). Furthermore, PeFA$^{Ucn3}$ silencing did not affect motor activity and, importantly, did not reduce adult intruder-directed aggression (*Figure 2G*). These

data suggest that PeFA$^{Ucn3}$ neuron activity is involved in the regulation of pup-directed aggression but not pup investigation, motor control, nor adult-mediated aggression.

## Activation of PeFA$^{Ucn3}$ neurons elicits infant-directed neglect in virgin females

To test if the activity of PeFA$^{Ucn3}$ neurons is sufficient to trigger infant-directed agonistic interactions in females, we used conditional viruses to express the excitatory opsin channelrhodopsin-2 (ChR2), or a control fluorophore, in PeFA$^{Ucn3}$ neurons (*Figure 3A*, *Figure 3—figure supplement 1A*). Parenting experience strongly impacts subsequent interactions of virgin females with pups (*Lonstein and De Vries, 2000*; *Okabe et al., 2013*), and indeed, sessions in mice injected with ChR2 in which the laser OFF condition was tested first –leading to an initial parenting bout – led to irreversible parental behavior in all subsequent sessions (*Figure 3—figure supplement 1B and C*). Hence, all sessions were started with the laser ON condition in both experimental and control groups, followed by laser OFF in the second session, and back to laser ON in third session (see Materials and methods). As expected, control females improved their parenting from session to session, with more pups retrieved and with shorter latencies in each subsequent session (*Figure 3B and C*). By contrast, optogenetic stimulation of PeFA$^{Ucn3}$ neurons reduced female parenting with fewer pups collected and with longer latencies in the third session compared to first (laser ON) and second (laser OFF) sessions (*Figure 3B and C*). Further, the duration of parental behaviors, investigation bout length, and time spent in the nest with pups were significantly reduced in the third compared to earlier sessions (*Figure 3C–E*), and fewer females in the ChR2 group retrieved pups to the nest (*Figure 3F*). While the overall display of pup-directed aggression (duration of aggression and number of females displaying aggression) was not significantly increased (*Figure 3G*), a subset of females (n = 3/15) showed tail-rattling locked to laser stimulation, an overt sign of aggression never observed in control females (*Video 1*). Further, activation of PeFA$^{Ucn3}$ neurons did not affect motor activity or intruder-directed aggression (*Figure 3H*). Consumption of appetitive food was slightly decreased with activation of PeFA$^{Ucn3}$ neurons in ChR2 females (*Figure 3—figure supplement 1D*), in agreement with the previously reported anorexigenic effect of this neural population (*Fekete et al., 2007*; *Jamieson et al., 2011*). To validate the efficiency of ChR2-mediated excitation of PeFA$^{Ucn3}$ neurons, we used in vitro slice electrophysiology and performed whole-cell patch clamp recordings from mCherry-positive PeFA$^{Ucn3}$ neurons (*Figure 3—figure supplement 1E*) to test various stimulation paradigms (465 nm blue light pulses of 5–20 ms pulse width at 2–20 Hz, *Figure 3—figure supplement 1F and G*).

To further confirm the effects of PeFA$^{Ucn3}$ neuron activation and assess the effects of prolonged activation in virgin females, we injected a conditional virus expressing the excitatory chemogenetic effector hM3Dq into either *Ucn3*::Cre+ or control Cre- females, followed by administration of its ligand clozapine-*n*-oxide (CNO) (*Figure 4—figure supplement 1A*). To test the efficacy of chemogenetic activation of *Ucn3*::Cre+ neurons, we used in vitro slice electrophysiology and performed whole-cell patch recordings from PeFA$^{Ucn3}$ neurons (*Figure 4—figure supplement 1C–E*) labeled with mCherry (*Ucn3*::Cre male mice injected with pAAV-hSyn-DIO-hM3D(Gq)-mCherry). Bath application of 10 μM CNO resulted in a significant increase in the firing rate of the PeFA$^{Ucn3}$ neurons (*Figure 4—figure supplement 1E*, p=0.0013 compared to baseline, Wilcoxon rank-sum method).

In vivo chemogenetic activation of PeFA$^{Ucn3}$ neurons led to a decrease in pup retrieval compared to control (Cre-) females (18% pups retrieved versus 70%, Fisher's exact test p<0.0015) (*Figure 4B*; Kolmogorov–Smirnoff test p<0.001). Further, fewer experimental females displayed parental behaviors compared to controls, and we observed a decrease in overall parental behavior, reduced time in the nest with pups, and reduced nest building (*Figure 4C and D*). Pup-directed aggression was observed in a few females after activation of PeFA$^{Ucn3}$ neurons, including infant-directed attack and aggressive carrying (*Figure 4E*; see Materials and methods). In addition, despite diminished parenting, females with activated PeFA$^{Ucn3}$ neurons investigated pups as much as control females, suggesting that they were not avoidant (*Figure 4E*). Together, these data suggest that activation of PeFA$^{Ucn3}$ neurons reduced parenting in virgin females, and in a few cases induced signs of aggression. Of note, pilot attempts to stimulate PeFA$^{Ucn3}$ neurons in mothers and to inhibit PeFA$^{Ucn3}$ neurons in fathers showed no behavioral effects, suggesting that mated animals were insensitive to these manipulations, which were thus not pursued further (see Materials and methods).

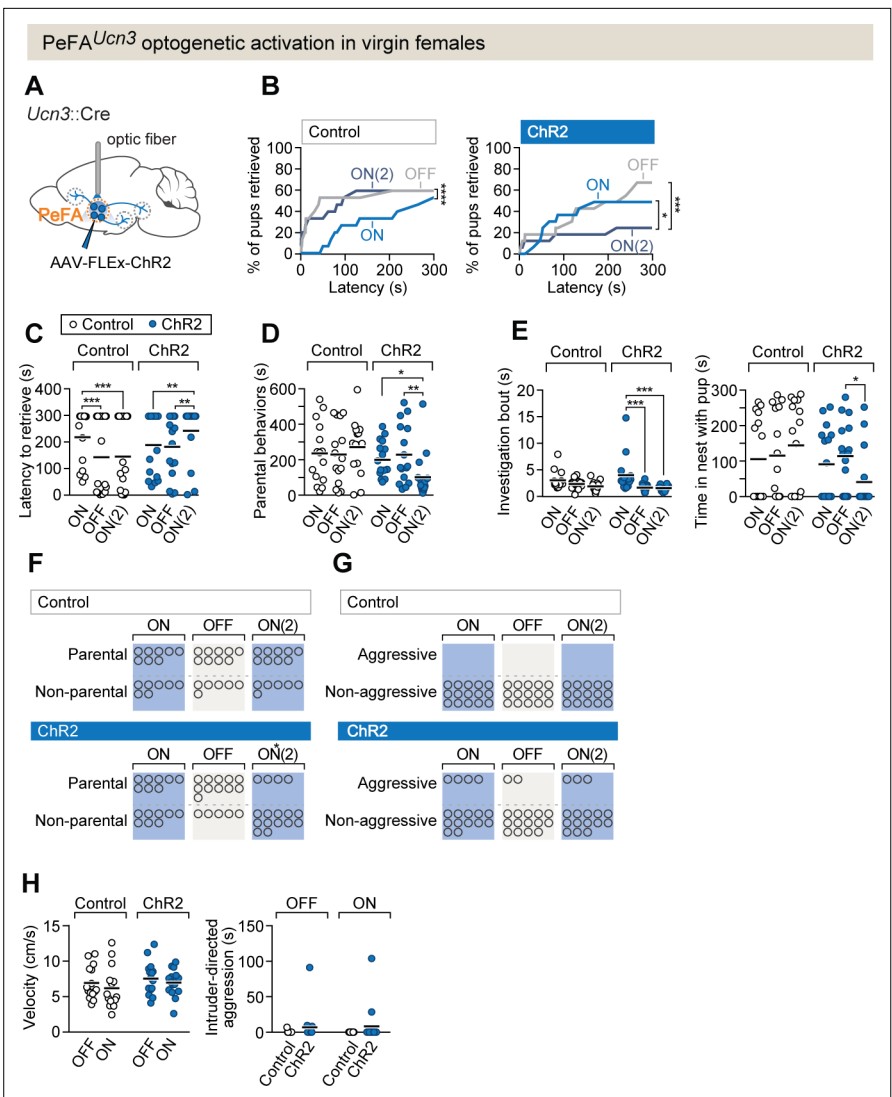

**Figure 3.** PeFA$^{Ucn3}$ optogenetic activation in virgin females suppresses parental behavior. (**A**) Setup for optogenetic activation of PeFA$^{Ucn3}$ neurons in virgin female mice. (**B**, left) Cumulative retrieval of pups in virgin females injected with a control virus (n = 15). Friedman statistic$_{(3, 27)}$ = 41.84 followed by Dunn's multiple comparisons tests. (Right) Cumulative retrieval of pups in virgin females injected with conditional virus expressing ChR2 (n = 16). Friedman statistic $_{(3, 24)}$ = 17.19, followed by Dunn's multiple comparisons test. (**C**) Latency to retrieve in control (open dots) and ChR2-expressing (blue dots) females across three sessions. Two-way repeated measures ANOVA reveals no main effect of virus treatment $F_{1-29}$ = 0.7892, but significant effects of session $F_{2,58}$ = 6.181 (p=0.0037) and interaction $F_{2,58}$ = 13.42 (p<0.0001). Post-hoc analysis by Bonferroni's multiple comparisons. (**D**) Duration of parental behaviors displayed by control (open dots) and ChR2-expressing (blue dots) females. Two-way repeated measures ANOVA reveals no main effect of virus $F_{1-29}$ = 2.721 or session $F_{2,58}$ = 1.178, but a significant main interaction effect $F_{2,58}$ = 4.747 (p=0.0123) followed by Tukey's post-hoc analysis. (**E**, left) Duration of time spent investigating pups in a single bout across three sessions in control (open dots) versus ChR2 (blue dots) females. Two-way repeated measures ANOVA reveals no significant main effect of virus treatment $F_{1-29}$ = 1.637 or session $F_{2,58}$ = 0.6324, but a significant interaction effect $F_{2,58}$ = 3.746 (p=0.0295). Post-hoc analysis by Tukey's multiple comparisons test. (Right) Time spent in the nest with pups. Two-way repeated measures ANOVA shows a significant main interaction effect $F_{2,58}$ = 3.746 (p<0.03). Post-hoc analysis by Tukey's multiple comparisons test. (**F**) Number of females displaying parental behaviors across three sessions in control and ChR2 females (accelerated failure regression model with first session as time-point baseline and control as treatment baseline, significant difference between control and ChR2 females in second laser ON session p<0.04). (**G**) Number of females in control or ChR2 groups displaying aggressive behaviors toward pups. (**H**, left) Locomotion velocity in control or ChR2 females with or without laser stimulation is not significantly affected. (Right) Aggression toward an adult male intruder is unaffected.

*Figure 3 continued on next page*

*Figure 3 continued*

The online version of this article includes the following figure supplement(s) for figure 3:

**Figure supplement 1.** Optogenetic stimulation of PeFA urocortin-3 expressing neurons leads to behavioral and physiological effects.

## PeFA<sup>Ucn3</sup> neurons receive input from parenting and stress-associated cell populations

To identify monosynaptic inputs to PeFA<sup>Ucn3</sup> neurons, we used conditional rabies virus-mediated retrograde trans-synaptic tracing in adult virgin males and females (*Wickersham et al., 2007*; *Figure 5A*). Fifteen brain areas were identified as providing monosynaptic inputs to PeFA<sup>Ucn3</sup> neurons, which, strikingly, all resided within the hypothalamus, lateral septum, and medial amygdala, with the highest fraction of inputs originating from PVH and MPOA (*Figure 5B–D*). In particular, the PVH contained the highest fractional inputs to PeFA<sup>Ucn3</sup> neurons in both sexes, with a higher fraction in females (44.92% ± 8.1%) compared to males (22.95% ± 4.2%) (*Figure 5D*). Neuronal inputs originating from the PVH were diverse and included Avp (31.9% ± 11.1%), *Crh* (17.8% ± 7.1%), and Oxt (9.1% ± 2.1%)-expressing cells, that is, populations implicated in social and stress responses (*Swanson and Sawchenko, 1980*; *Jezova et al., 1995*; *Donaldson and Young, 2008*; *Insel, 2010*). In the MPOA, which constituted the second major input area to PeFA<sup>Ucn3</sup> neurons, around half (48.02%) of PeFA<sup>Ucn3</sup> monosynaptic input were Gal+ cells, a largely GABAergic neural population involved in the positive control of parenting (*Wu et al., 2014*; *Kohl et al., 2018*; *Figure 5E and F*). Thus, PeFA<sup>Ucn3</sup> neurons are poised to collect information from vomeronasal and hypothalamic pathways associated with stress and social behavior control, including inhibitory input from parenting behavior circuits.

## PeFA<sup>Ucn3</sup> neurons project to areas involved in aggression and anxiety

Next, we visualized PeFA<sup>Ucn3</sup> axons and presynaptic terminals by conditional viral expression of synaptophysin-eGFP and tdTomato (*Figure 6A*). All observed PeFA<sup>Ucn3</sup> projections were within the hypothalamus and limbic areas, with the most densely labeled projections in AHi, LSv, PVH, and VMH (*Figure 6B–D*). No obvious sex differences were observed (*Figure 6B–D*).

To better understand the organization of PeFA<sup>Ucn3</sup> neuronal projections, we performed pairwise injections of the fluorophore-conjugated cholera toxin B subunit (CTB-A488 and CTB-A647) retrograde tracers (*Figure 7A and B*). We focused on PeFA<sup>Ucn3</sup> projections to LS, AHi, and VMH, based on their dense labelling and on the known functions of these target areas in stress and aggression (*Slotnick et al., 1973*; *Blanchard et al., 1977*; *Albert and Chew, 1980*; *Kruk et al., 1983*; *Canteras et al., 1997*; *Lin et al., 2011*; *Anthony et al., 2014*; *Lee et al., 2014*). For each paired target injection, around 50–60% of PeFA<sup>Ucn3</sup> neurons were labeled by both tracers (*Figure 7C*), suggesting that individual PeFA<sup>Ucn3</sup> neurons project to multiple targets. To further determine the respective organization of these projections from the PeFA and their involvement in pup-directed aggression, we performed CTB-mediated retrograde tracing from LS, VMH, and AHi individually and quantified the overlap between CTB labeling and Fos expression in the PeFA of virgin males after infant-directed attack (*Figure 7D and E*). Fos labeling was found in PeFA neurons associated with each of the three major projections, with AHi-projecting Fos+ cells preferably located in the rostral and medial subdivision of PeFA (*Figure 7F and G*).

## Functional analysis reveals strongest connectivity between PeFA<sup>Ucn3</sup> neurons and AHi

To assess the strength of the functional connectivity between PeFA<sup>Ucn3</sup> neurons and downstream targets, we used conditional AAVs to

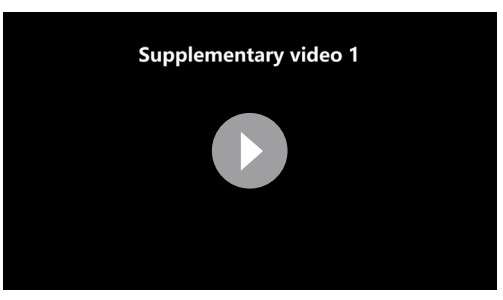

**Video 1.** Optogenetic stimulation of PeFA<sup>Ucn3</sup> neurons in a virgin female mouse during pup-directed behavior. A newborn pup is presented to a female in the home cage. Optogenetic stimulation is indicated by the white dot in the upper center of the frame. In a subset of females, we observe tail rattling behavior toward the pup during light illumination as seen in this example.
https://elifesciences.org/articles/64680/figures#video1

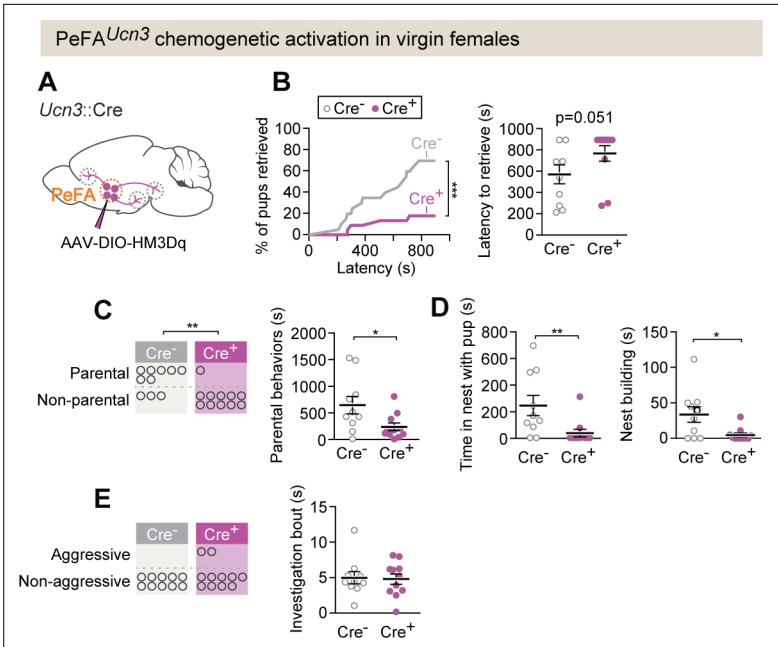

**Figure 4.** PeFA$^{Ucn3}$ chemogenetic activation in virgin females suppresses parental behavior. (**A**) Setup for chemogenetic activation of PeFA$^{Ucn3}$ neurons in virgin females. (**B**, left) Cumulative retrieval in Cre+ (n = 11) or Cre- control (n = 10) females. Kolmogorov–Smirnov test for significance p<0.0001. (Right) Latency to retrieve first pup, Mann–Whitney test p=0.051. (**C**, left) Number of females in Cre + or Cre- (control) displaying parental behaviors. Significance determined by Fisher's exact test (p=0.0075). (Right) Duration of parental behaviors, t-test p<0.0291. (**D**, left) Duration in nest, Mann–Whitney test, p=0.0039. (Right) Nest building, Mann–Whitney test p=0.0422. (**E**, left) Number of females in Cre + or Cre - (control) displaying infant-directed aggression. (Right) Investigation bout length is unaffected.

The online version of this article includes the following figure supplement(s) for figure 4:

**Figure supplement 1.** Chemogenetic stimulation of PeFA urocortin-3 expressing neurons significantly increases cell firing rate.

express ChR2 and mCherry in PeFA$^{Ucn3}$ neurons (*Figure 8A and B*). We performed whole-cell patch clamp recordings from putative postsynaptic neurons in the three targeted areas: LS, VMH, and AHi (*Figure 8A and B*). Our data showed that PeFA$^{Ucn3}$ neurons make functional connections with neurons in AHi with a mean-evoked excitatory post-synaptic currents (EPSC) amplitude of 47.32 ± 4.8 pA (four out of seven neurons), in LS with a mean-evoked amplitude of 20.78 ± 1.42 pA (seven out of seven neurons) and in VMH with a mean-evoked amplitude of 9.75 ± 2.67 pA (two out of nine neurons) (*Figure 8C and D*). Based on the observed evoked EPSC amplitudes, the AHi appears to receive significantly stronger connections from PeFA$^{Ucn3}$ (*Figure 8D*) than the other two areas tested (p=2.55 * 10$^{-7}$ for AHi to LS comparison and p=2.73 * 10$^{-8}$ AHi to VMH comparison, Wilcoxon rank-sum test).

Next, we used a paired pulse stimulation protocol to estimate the probability of connectivity and probability of release from PeFA$^{Ucn3}$ neurons to each of these three brain areas (*Figure 8E*). We found that PeFA$^{Ucn3}$ projections to LS and AHi showed high probability of release (*Figure 8E*, top and bottom), as evident from the significant depression to the paired pulse paradigm (p=2.10 * 10$^{-5}$ for AHi and p=2.20 * 10$^{-3}$ for LS for paired pulse depression, Wilcoxon rank-sum test). By contrast, PeFA$^{Ucn3}$ projections to VMH displayed weak probability of release (p=4.3 * 10$^{-2}$ for paired pulse depression, Wilcoxon rank-sum test) as evidenced by lack of depression to the paired pulse paradigm (*Figure 8E*, middle row).

These functional connectivity data demonstrate that PeFA$^{Ucn3}$ projections to AHi make significantly stronger connections (*Figure 8D*) with higher probability of release (*Figure 8E*, bottom) compared to PeFA$^{Ucn3}$ projections to other brain areas (i.e., LH and VMH, *Figure 8D and E*, top and middle). Together with our anatomical data, these results suggest that all major projections are active during pup-directed attacks with preferential involvement of rostro-medial PeFA$^{Ucn3}$ projections to AHi.

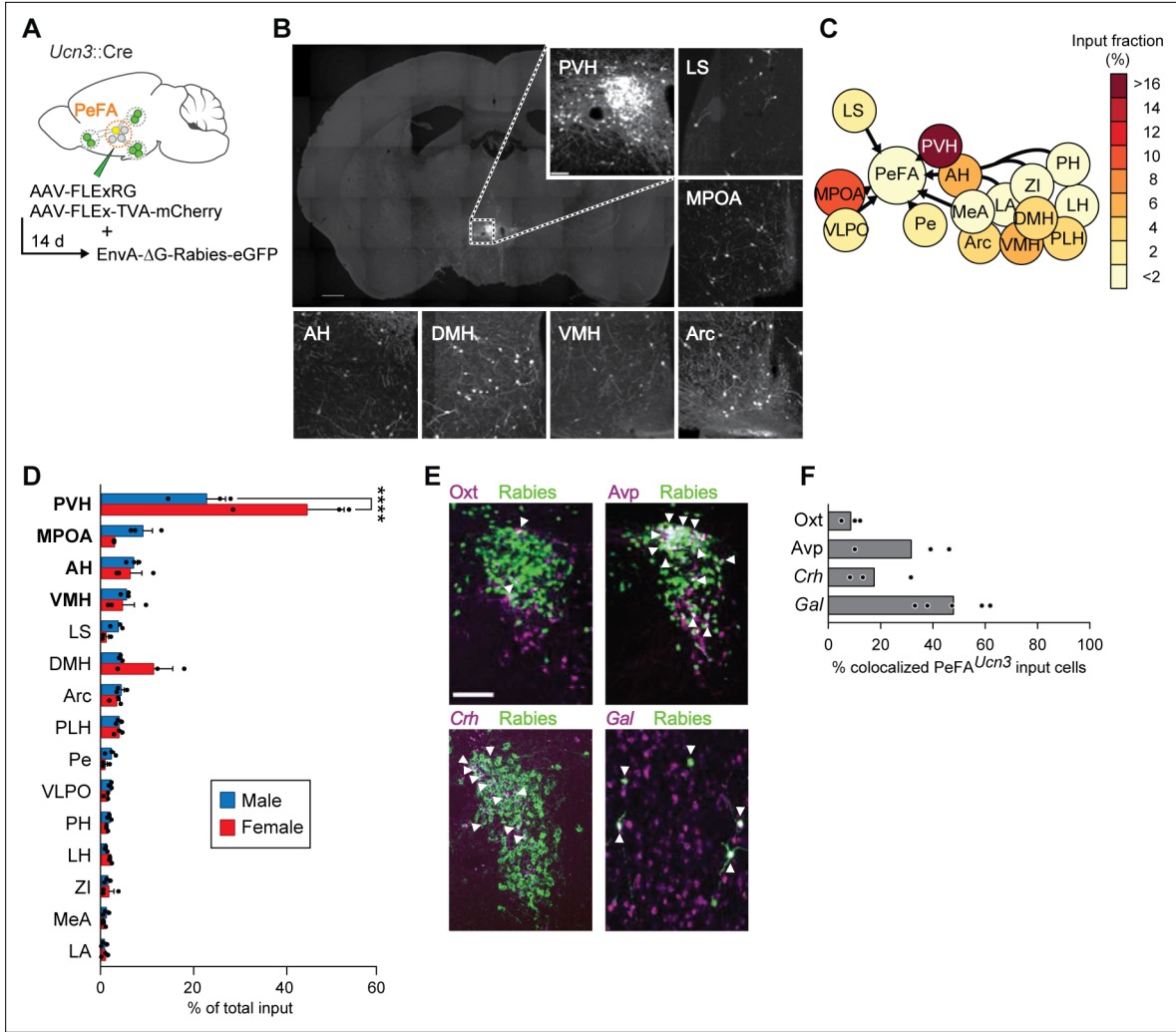

**Figure 5.** Inputs of PeFA[Ucn3] neurons. (**A**) Schematic of monosynaptic retrograde tracing from PeFA[Ucn3] neurons. (**B**) Representative images of labeled monosynaptic inputs (scale bar 200 μm). (**C**) Heatmap of input fractions by brain area in virgin males. (**D**) Quantification of input fraction in virgin males (n = 3) and virgin females (n = 3). Regions are ranked by effect size largest (top) to smallest (bottom) relative to values observed in males using a regression analysis. Paraventricular hypothalamus (PVH) and medial preoptic area (MPOA), and to a lesser extent amygdalohippocampal area (AH) and ventromedial hypothalamus (VMH), showed significantly enriched input to the PeFA[Ucn3] neurons compared to chance (p<0.0001, indicated in bold), post-hoc test significant for PVH male vs. female. (**E**) Representative images of immunofluorescence (Oxt, Avp) or in situ hybridization (corticotropin-releasing hormone *Crh*, *Gal*) co-staining with Rabies-eGFP (scale bar, 200 μm), arrowheads indicate colocalization. (**F**) Percentages of presynaptic eGFP-positive neurons expressing Oxt, Avp, *Crh*, or *Gal* (n = 3–5/group). MPOA input neurons largely colocalize with galanin expression (48.02% ± 5.63%).

## PeFA[Ucn3] neuronal projections govern discrete aspects of infant-directed neglect and aggression

To test the respective functions of PeFA[Ucn3] targets, we performed optogenetic activation of PeFA[Ucn3] terminals in virgin females injected in the PeFA with a conditional virus expressing ChR2 and with optical fibers implanted bilaterally above each projection target (*Figure 9*). We initially focused on projections to VMH, based on the documented role of this area in the control of aggression (*Kruk et al., 1983*; *Lin et al., 2011*). Stimulation of PeFA[Ucn3] to VMH projections reduced pup retrieval across sessions and shortened individual bouts of pup investigation (*Figure 9B*, *Video 2*). However, it did not affect the overall number of parental females, nor the overall time engaged in parental interactions, latency to retrieve, time spent in the nest with pups, or pup grooming (*Figure 9C–E*). Further, we did not observe any pup-directed aggression (*Figure 9F*). Slice recordings (*Figure 8*) had indicated that both the probability and the strength of connectivity between PeFA[Ucn3] and VMH neurons are low, perhaps in part explaining the weak behavioral manifestation of the selective activation of PeFA[Ucn3] to

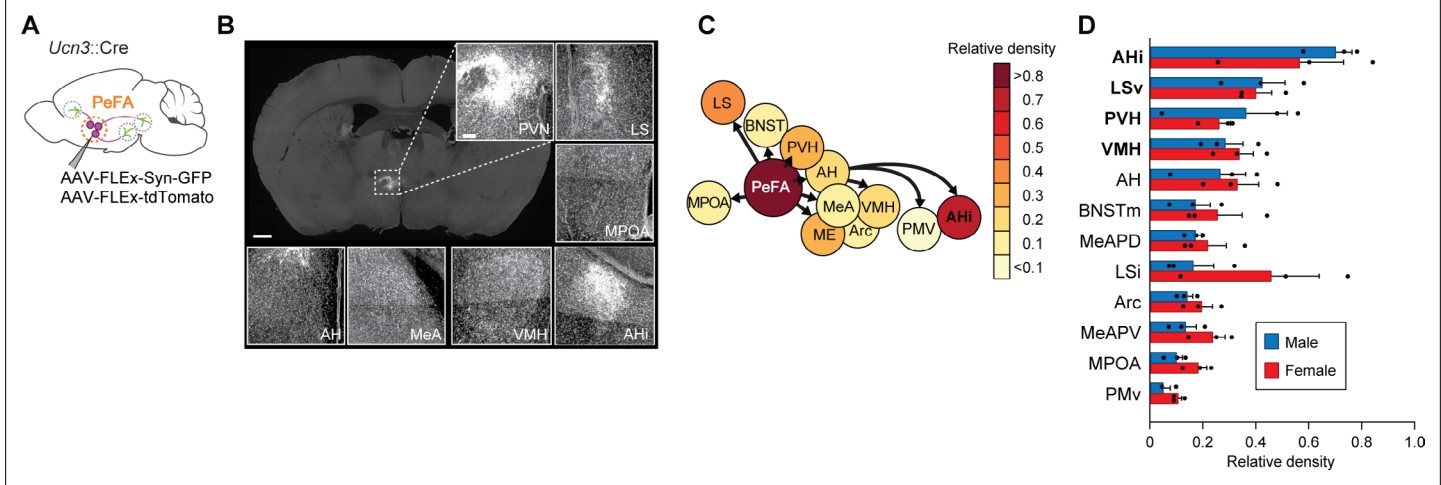

**Figure 6.** Projections of PeFA$^{Ucn3}$ neurons. (**A**) Anterograde tracing strategy. (**B**) Representative images of synaptophysin densities in projection areas (scale bar 200 μm). (**C**) Heatmap of relative densities of PeFA$^{Ucn3}$ projections. (**D**) Quantification of relative densities of PeFA$^{Ucn3}$ projections in males (n = 3) and females (n = 3), regression analysis reveals significant enrichment of amygdalohippocampal area (AHi), ventral lateral septum (LSv), paraventricular hypothalamus (PVH), and ventromedial hypothalamus (VMH) projections compared to chance (indicated in bold).

VMH projections. Thus, PeFA$^{Ucn3}$ to VMH projections inhibit sustained interactions with pups and pup retrieval but do not elicit aggression toward pups.

We next stimulated the PeFA$^{Ucn3}$ to LS projections (**Figure 9G**) and found that this manipulation inhibited parental behavior in a significant fraction of females and reduced cumulative pup retrieval and pup grooming across the three sessions (**Figure 9H and I**). Moreover, stimulation of PeFA$^{Ucn3}$ to LS projections triggered instances of pup-directed aggression (such as aggressive handling and biting) in 30% of females, although the overall time displaying aggression remained negligible (**Figure 9J**). PeFA$^{Ucn3}$ to LS stimulation had no significant effect on time spent in the nest with pups, pup investigation, overall parenting time, and latency to retrieve (**Figure 9K and L**). In a subset of females (n = 2/10), PeFA$^{Ucn3}$ to LS stimulation led to an escape response, which was never observed in other manipulations (**Video 3**). Together, these results suggest that PeFA$^{Ucn3}$ to LS projections inhibit parenting and mediate signs of repulsion and aggression.

Slice recordings revealed strong synaptic connectivity between PeFA$^{Ucn3}$ projections and AHi neurons (**Figure 8**), suggesting that the PeFA$^{Ucn3}$ to AHi projection may have a strong impact on pup-directed behaviors. Indeed, when we examined the function of PeFA$^{Ucn3}$ to AHi projections (**Figure 9M**), we found that 55% of females displayed infant-directed aggression such as biting and aggressive carrying of pups without completing retrieval, in a manner similar to the stereotyped infanticidal behaviors observed in males, with increased time displaying aggression (**Figure 9N**, **Video 4**). Additionally, PeFA$^{Ucn3}$ to AHi stimulation dramatically reduced cumulative pup retrieval over the course of three sessions, time spent in the nest with pups, number of parental females, and time spent pup-grooming (**Figure 9O and P**) with no effect on retrieval latency, pup investigation bout length, and time spent parenting (**Figure 9Q and R**). Thus, stimulation of PeFA$^{Ucn3}$ to AHi projections strongly increased the expression of infant-directed aggression and suppressed the expression of parental behavior.

In control experiments, we found that stimulation of PeFA$^{Ucn3}$ projections to VMH, LS, and AHi did not impact locomotion, appetitive feeding, or conspecific aggression (**Figure 9—figure supplement 1A–J**).

Altogether, these data suggest that various aspects and degrees of infant-directed neglect and aggression are mediated across PeFA$^{Ucn3}$ projections, with PeFA$^{Ucn3}$ to VMH projections suppressing pup investigation, PeFA$^{Ucn3}$ to LS projections mediating reduced pup handling and low-level aggression, and PeFA$^{Ucn3}$ to AHi projections inducing stereotyped displays of pup-directed aggression (**Figure 10**).

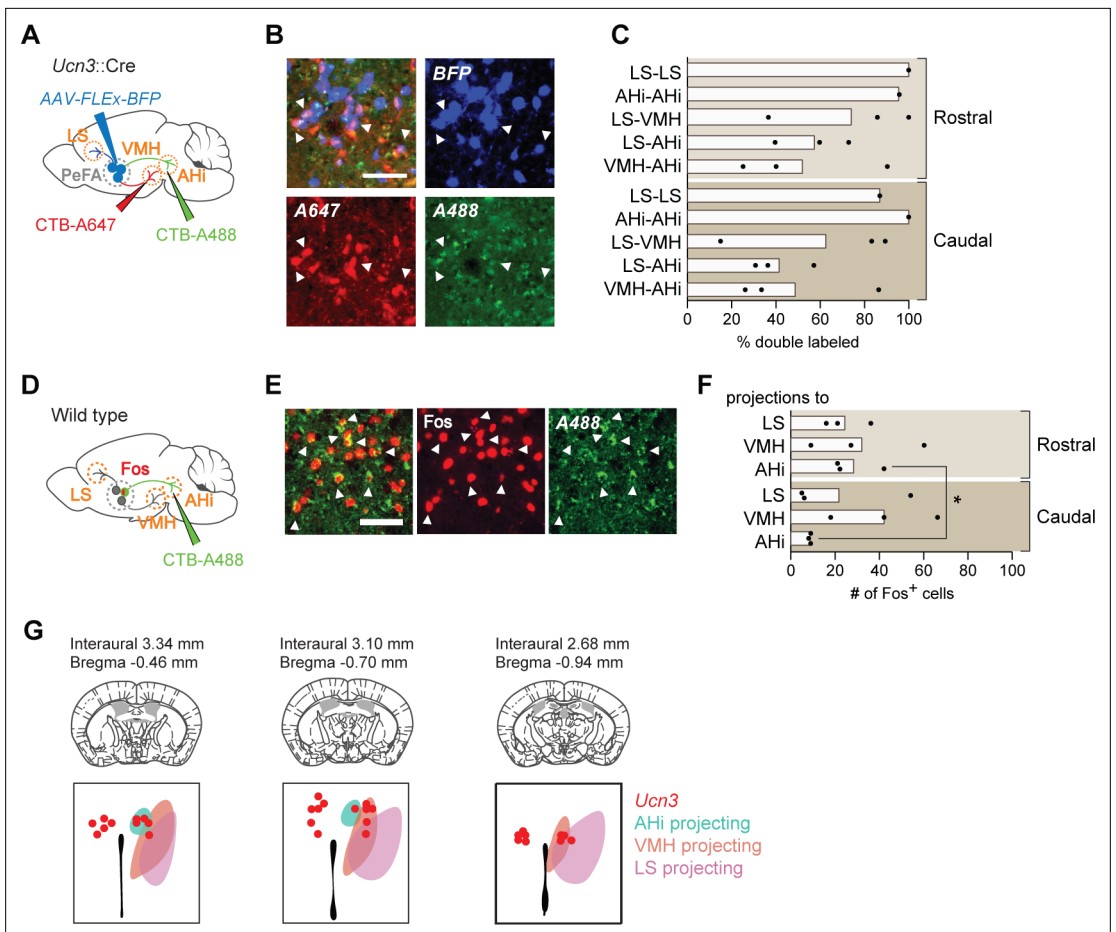

**Figure 7.** Projection tracing reveals bifurcating but behaviorally distinguishable sub-populations of PeFA[Ucn3] neurons. (**A**) Schematic of retrograde cholera toxin B (CTB) tracing combined with PeFA[Ucn3] reporter. (**B**) Representative image of CTB experiment (scale bar 100 μm). (**C**) Quantification of Ucn3+ neurons labeled with CTB. (**D**) Schematic of CTB tracing combined with Fos staining in infanticidal males. (**E**) Representative image of CTB and Fos labeling in periformical area (PeFA) after CTB injection in amygdalohippocampal area (AHi). (**F**) Quantification of number of Fos-positive cells labeled with CTB in the rostral and caudal PeFA after CTB injection in a given projection (n = 3/group). Rostral versus caudal percentages are significantly different for AHi-projecting neurons (paired t-test p<0.03). (**G**) Schematic of topographical arrangement of PeFA[Ucn3] neurons in PeFA according to their projection sites.

## Discussion

The last few decades have seen considerable progress in the use of experimental model systems to identify the neural basis of parental behavior (*Noirot, 1972*; *Fleming and Rosenblatt, 1974a*; *vom Saal, 1985*; *Lonstein and De Vries, 2000*; *Dulac et al., 2014*; *Wu et al., 2014*; *Marlin et al., 2015*; *Scott et al., 2015*; *Kohl et al., 2017*; *Kohl et al., 2018*; *Numan, 2020*; *Stagkourakis et al., 2020*). These studies have uncovered critical sensory cues underlying parent-infant interactions, as well as brain circuits underlying specific nurturing displays in males and females (*Dulac et al., 2014*; *Wu et al., 2014*; *Marlin et al., 2015*; *Scott et al., 2015*; *Kohl et al., 2017*; *Kohl et al., 2018*; *Yoshihara et al., 2021*). By contrast, the neural control of infant-directed aggression remains less understood. Initially interpreted as a pathological behavior, Sarah Hrdy's pioneering work in langurs, as well as work in other mammalian species, instead put forward infanticide as an adaptive behavior that enables males to gain advantage over rivals and females to adapt reproduction to adverse circumstances (*Hrdy, 1974*; *Hrdy, 1979*). Although a number of brain areas have been described that negatively affect parental behavior (*Fleming and Rosenblatt, 1974a*; *Fleming and Rosenblatt, 1974b*; *Fleming et al., 1980*; *Gammie and Nelson, 2001*; *Amano et al., 2017*; *Isogai et al., 2018*), the existence of a dedicated neural circuitry controlling infant-directed agonistic interactions, including infanticide, had not yet been explored.

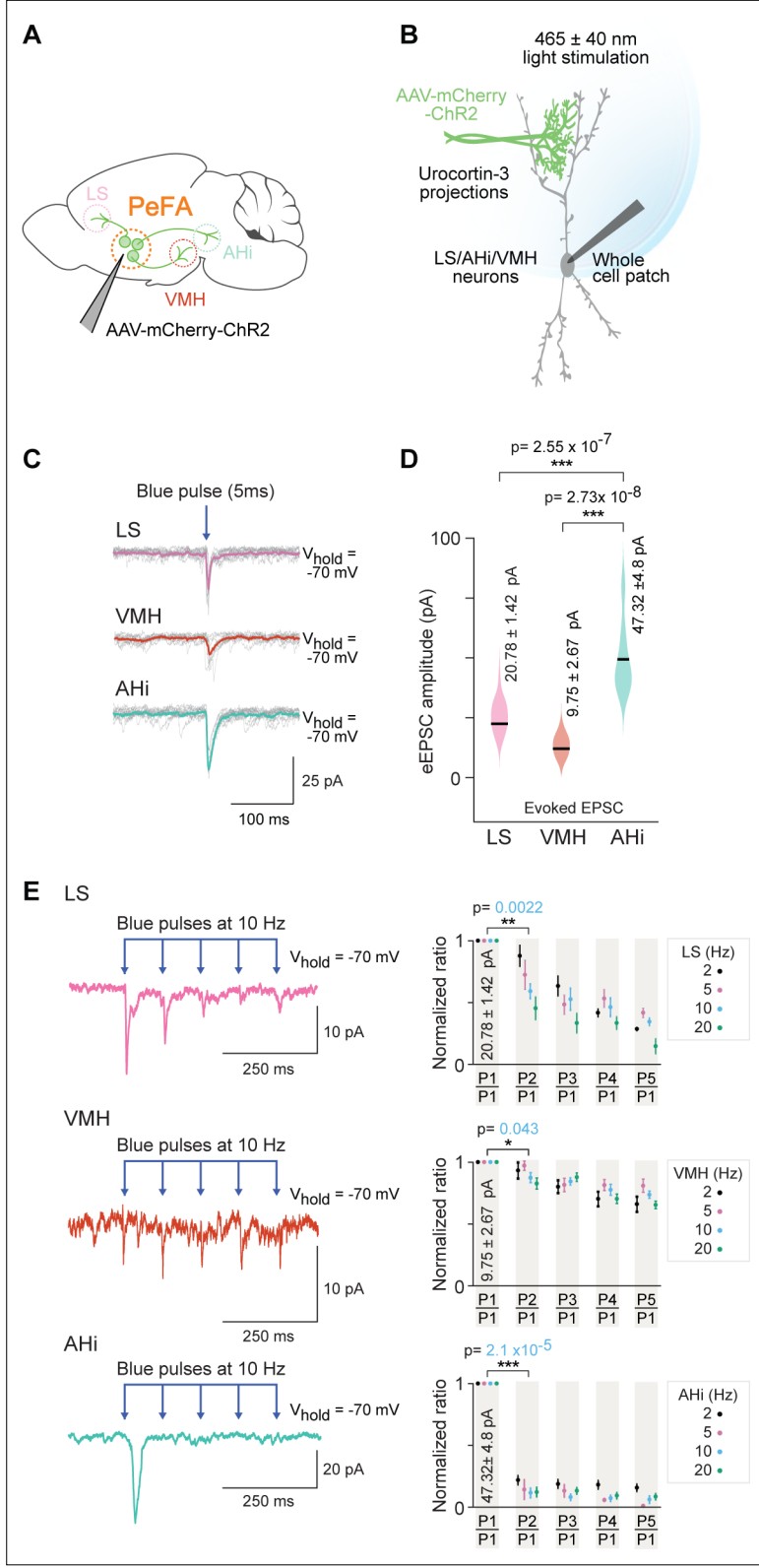

**Figure 8.** Optogenetic investigation of functional connectivity between PeFA[Ucn3] neurons and specific targets. (**A**) Experimental design depicting selective targeting of PeFA[Ucn3] with ChR2 (Ucn3::Cre injected with AAV1-eF1a-DoubleFlox hChR2(H134R)-mCherry-WPRE-HgHpA, 8–12- week-old male mice, n = 8, five males and three females). (**B**) Schematic depicting in vitro slice electrophysiological recordings of neurons in lateral septum (LS),

*Figure 8 continued on next page*

*Figure 8 continued*

ventromedial hypothalamus (VMH), and amygdalohippocampal area (AHi) slices in response to optogenetic activation of PeFA$^{Ucn3}$ projections labeled with ChR2. (**C**) Electrophysiological responses to activation of PeFA$^{Ucn3}$ projections (single light pulse: 5 ms duration, 465 nm at 5 mW/mm$^2$) for 10 trials with average traces shown in pink (LS), orange (VMH), and green (AHi). (**D**) Summary plot showing significantly higher amplitude of evoked excitatory post-synaptic currents (eEPSCs) in neuronal populations of AHi (green) compared to LS (pink) (Wilcoxon rank-sum test, p<0.0001) and VMH (orange) (Wilcoxon rank-sum test, p<0.0001). (**E**) Electrophysiological trace of an LS neuron (top left), a VMH neuron (middle left), and an AHi neuron (bottom left) (mean response of five consecutive trials) depicting paired pulse depression of EPSCs in response to activation of PeFA$^{Ucn3}$ projections (light pulses: 5 ms duration, 465 nm at 10 Hz). Summary plots showing paired pulse depression of optogenetic activation of PeFA$^{Ucn3}$ projections on LS neurons (top right, Wilcoxon rank-sum test, p=0.0022), VMH neurons (middle right, Wilcoxon rank-sum test, p=0.043), and AHi neurons (bottom right, Wilcoxon rank-sum test, p<0.001) as a function of stimulation frequency.

In the present study, we demonstrate that PeFA$^{Ucn3}$ neurons are specifically activated by pup-directed aggression in males and females, but not by feeding, adult conspecific aggression, or acute restraint stress. Through functional manipulations, we find that PeFA$^{Ucn3}$ neurons are required for infant-directed aggression in males. In females, direct activation of PeFA$^{Ucn3}$ neuronal projections to AHi elicited overt aggression toward pups while somatic activation of PeFA$^{Ucn3}$ neurons strongly reduced maternal behavior including pup retrieval, time in the nest, and overall parenting time but did not cause widespread aggressive displays, except tail rattling in a few animals. Our input analysis of PeFA$^{Ucn3}$ neurons provided us with useful information to understand the different outcomes obtained by manipulation of PeFA neurons in males and females. Indeed, our study identified MPOA$^{Gal}$ neurons, an inhibitory neuronal population driving the positive control of parenting (***Wu et al., 2014***; ***Kohl et al., 2018***), as providing major inputs to PeFA$^{Ucn3}$ neurons. MPOA$^{Gal}$ neurons were also shown to share common targets with PeFA$^{Ucn3}$ neurons, such as AHi, LS, and VMH (***Kohl et al., 2018***). Activation of MPOA$^{Gal}$ neurons has been shown to robustly, reversibly, and acutely enhance parental behavior in virgin females and fathers and to suppress ongoing infant-directed aggression in virgin males (***Wu et al., 2014***; ***Kohl et al., 2018***). In our experiments, MPOA$^{Gal}$ neurons are likely to be strongly activated in females exposed to pups, thus presumably exerting their putative inhibitory role on PeFA$^{Ucn3}$ neurons and their projections, therefore occluding the full expression of pup-directed agonistic behavior following artificial activation of PeFA$^{Ucn3}$ neurons in females but less so when more direct activation is provided to PeFA$^{Ucn3}$ neuronal projections to AHi. Further investigation of PeFA$^{Ucn3}$ neuron function while MPOA$^{Gal}$ neuron activity is inhibited will be necessary to address this hypothesis.

Inhibition of PeFA$^{Ucn3}$ neurons has a robust effect to suppress ongoing infant-directed attack behavior, but unlike MPOA$^{Gal}$ neuron stimulation, this effect appears long-lasting and does not reverse in the short term. We interpret this result as indicating that suppression of PeFA$^{Ucn3}$ neuron activity induces downstream circuit plasticity that favors the stable expression of parenting and inhibition of pup-mediated aggression. Indeed, we attempted to activate PeFA$^{Ucn3}$ neurons in mothers and fathers in pilot experiments and were unable to observe increased neglect or aggression, suggesting that PeFA$^{Ucn3}$ neuron activation cannot overcome the MPOA$^{Gal}$ circuit activity in parental animals.

Previous studies have shown that electrical stimulation of the PeFA leads to a so-called 'hypothalamic rage response' comprising typical aggressive displays including piloerection, teeth baring, and bite attacks (***Slotnick et al., 1973***; ***Kruk et al., 1983***; ***Canteras et al., 1997***). Indeed, we observe *Fos*+ neurons in the PVH and PeFA of males and females displaying conspecific aggression, but these neurons are *Ucn3*-negative, suggesting that distinct PeFA populations mediate aggression toward either adults or infants. In addition, previous studies have implicated *Ucn3* in the PeFA in the regulation of social recognition, stress responsivity, and, more recently, risk assessment (***Venihaki et al., 2004***; ***Deussing et al., 2010***; ***Kuperman et al., 2010***; ***Jamieson et al., 2011***; ***Horii-Hayashi et al., 2021***). However, our experiments with targeted inhibition or stimulation of PeFA$^{Ucn3}$ neurons specifically affected parental behavior with no impact on adult social behavior, indicating that PeFA$^{Ucn3}$ neurons may be specifically tuned to social cues from infants and/or juveniles rather than adult conspecifics.

Specific tuning of these neurons to social cues is supported by our finding that input areas to PeFA$^{Ucn3}$ neurons include the MPOA, MeA, and PVH, known to be essential for integration of social cues sensed by the vomeronasal system that has been well-established as a crucial pathway for infanticide (***Tachikawa et al., 2013***; ***Wu et al., 2014***). MPOA$^{Gal}$ neurons directly send input to PeFA$^{Ucn3}$

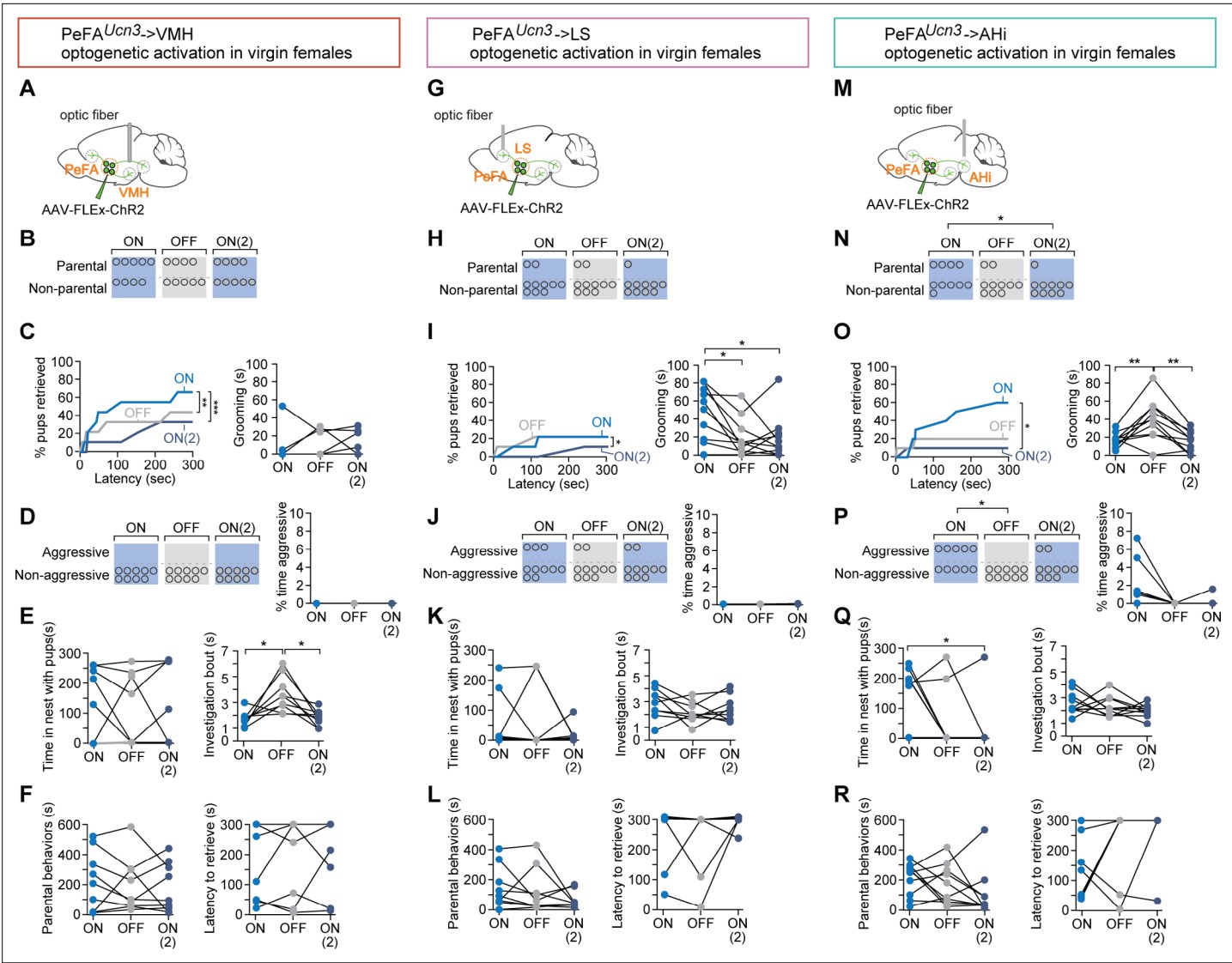

**Figure 9.** Specific projections from PeFA$^{Ucn3}$ neurons mediate infant-directed neglect and aggression. (**A**) Optogenetic stimulation of PeFA$^{Ucn3}$ to ventromedial hypothalamus (VMH) fiber terminals. (**B**) Number of females showing parental behavior. (**C**, left) Cumulative retrieval of pups is significantly different among groups (Friedman statistic $_{(3-15)}$ = 19.57, p<0.0001; n = 9). Dunn's multiple comparisons test reveals difference between ON vs. ON2 (***) and OFF vs. ON2 (*). (Right) One-way repeated measures ANOVA revealed a significant main effect in investigation bout length $F_{(1, 11)}$ = 10.12 (p<0.005) among groups. Tukey's multiple comparisons revealed differences between ON vs. OFF (*) and OFF vs. ON2 (*). (**D**, left) Number of females showing pup-directed aggression and (right) percent time displaying aggression show no differences across sessions. (**E**, left) Time in the nest with pups and (right) grooming were not altered by PeFA$^{Ucn3}$ to VMH fiber stimulation. (**F**, left) Duration of parental behaviors and (right) latency to retrieve pups were not changed. (**G**) Optogenetic stimulation of PeFA$^{Ucn3}$ to lateral septum (LS) terminals. (**H**) Number of females showing parental behavior is not affected (n = 10). (**I**, left) Cumulative retrieval of pups is suppressed and significantly affected over the course of the experiment analyzed by Friedman test, Friedman statistic $_{(3-7)}$ = 10.21 (p<0.0041). Dunn's multiple comparisons test reveals significant differences in OFF vs. ON2 comparison (*). (Right) One-way repeated measures ANOVA reveals that grooming is significantly reduced over the course of the experiment $F_{2-16}$ = 6.368 (p<0.0102). Tukey's multiple comparisons test reveals significant differences between ON vs. OFF (*) and ON vs. ON2 (*). (**J**, left) Number of females showing aggression and (left) time spent aggressive toward pups is not changed. (**K**, left) Time in the nest and (right) investigation bout length are not affected. (**L**, left) Parental behaviors and (right) latency to retrieve are unaffected. (**M**) Schematic of optogenetic stimulation of PeFA$^{Ucn3}$ to amygdalohippocampal area (AHi) terminal stimulation. (**N**) Number of females showing parental behaviors was significant reduced between ON vs. ON2 conditions (chi-square p<0.0191). (Right) One-way repeated measures ANOVA revealed reversibly reduced grooming by stimulating of PeFA$^{Ucn3}$ to AHi axonal terminals $F_{2,18}$ = 11.40 (p<0.0006). Tukey's multiple comparisons test showed differences between ON vs. OFF (**) and OFF vs. ON2 (**). (**O**, left) Friedman test reveals that cumulative retrieval of pups is significantly affected (p<0.0095). Dunn's multiple comparisons shows significant differences between ON vs. ON2 (*). (Right) Time spent in the nest with pups was significantly reduced between ON vs. ON2 conditions (repeated measures ANOVA $F_{2,14}$ = 4.297, p<0.0351, followed by Tukey's post-hoc test). (**P**, left) Number of females showing aggressive behaviors toward pups is significantly different between laser ON and laser OFF groups (chi-square test p<0.02; n = 10). (Right) Friedman test reveals a significant difference in the time spent displaying aggressive behavior

*Figure 9 continued on next page*

*Figure 9 continued*

over the course of the experiment (p<0.0247). (**Q**, left) Time spent in the nest with pups was significantly reduced between ON vs. ON2 conditions (repeated measures ANOVA $F_{(2,14)}$ = 4.297, p<0.0351, followed by Tukey's post-hoc test) (right) and investigation bout length were unaffected. (**R**, left) Duration of parental behaviors and (right) latency to retrieve were unchanged across sessions.

The online version of this article includes the following figure supplement(s) for figure 9:

**Figure supplement 1.** Optogenetic stimulation of PeFA urocortin-3 expressing neuronal projections does not impact locomotion, feeding, or inter-adult defensive behaviors.

---

neurons, suggesting a direct relationship between this well-established hub for pro-parental behavior and the putative infant-directed aggression neuron population described here. Inputs from PVH largely express *Avp* and *Crh*, and *Avp*-expressing PVH neurons have previously been implicated in male-typical social behavior (*Donaldson and Young, 2008*; *Insel, 2010*), while *Crh*-expressing PVH neurons are essential mediators of the physiological stress response (*Vale et al., 1981*; *Ulrich-Lai and Herman, 2009*).

Previously identified hypothalamic neuronal populations critical for feeding (e.g., *Agrp*-expressing neurons in the arcuate nucleus) and parenting (MPOA^Gal neurons) have been shown to be organized in distinct pools, each largely projecting to a distinct downstream area (*Betley et al., 2013*; *Kohl et al., 2018*). In contrast, individual PeFA^Ucn3 neurons project to multiple targets, in a similar manner to VMH neurons extending collaterals to downstream defensive circuits (*Wang et al., 2015*). Interestingly, the three major PeFA^Ucn3 target areas VMH, LS, and AHi convey overlapping but distinct aspects and degrees of infant-directed neglect and aggression. Our recording data suggests that PeFA^Ucn3 projections make direct excitatory functional connections with the neuronal populations of LS, VMH, and AHi. Among these areas, PeFA^Ucn3 projections to AHi showed particularly strong connectivity and high probability of release, and when stimulated lead to overt pup-directed attack behaviors. By contrast, weaker connectivity was identified between PeFA^Ucn3 and LS and VMH neurons, which may in part explain the less robust behavioral manifestations of selective activation of PeFA^Ucn3 projections to LS and VMH (*Wittmann et al., 2009*; *Chen et al., 2011*; *Battagello et al., 2017*).

Taking the connectivity strength and functional behavior results altogether, our data suggest that each projection may participate in different aspects and degrees of agonistic responses to pups. Based on the role of VMH in conspecific aggression (*Lin et al., 2011*; *Yang et al., 2013*), we hypothesized a role for PeFA^Ucn3 to VMH projection in pup-directed attack. However, we found instead that this projection regulates pup investigation. The PeFA^Ucn3 to LS projections also led to sustained reductions in pup interactions, a finding that is consistent with a recent study showing that LS neurons expressing corticotrophin-releasing factor receptor 2 (CRFR2), the high-affinity receptor for *Ucn3*, may control persistent anxiety behavior (*Anthony et al., 2014*) and, in addition, led to incidences of pup-directed aggression. The function of the AHi is not well understood, although recent data suggest that it may send sexually dimorphic projections to MPOA^Gal neurons essential for parenting (*Kohl et al., 2018*), while projections from the AHi to

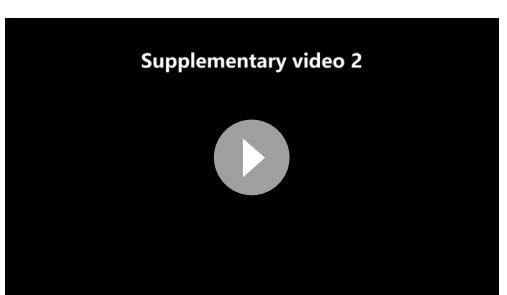

**Video 2.** Optogenetic stimulation of PeFA^Ucn3 neuronal projections to the ventromedial hypothalamus in a virgin female mouse during pup-directed behavior. Most females showed reduced interaction with the pup during light illumination as seen here.

https://elifesciences.org/articles/64680/figures#video2

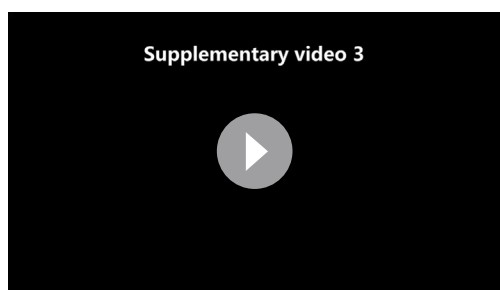

**Video 3.** Optogenetic stimulation of PeFA^Ucn3 neuronal projections to the lateral septum in a virgin female mouse during pup-directed behavior. Most females showed avoidance of the pup during light illumination, and in a subset of females we observed escape-like behavior upon light stimulation as in this example.

https://elifesciences.org/articles/64680/figures#video3

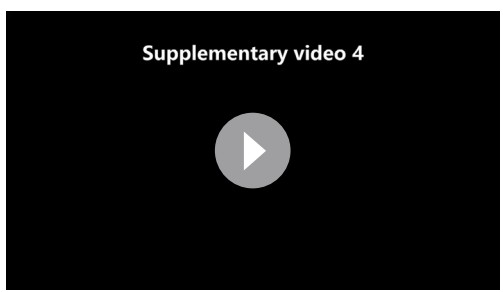

**Video 4.** Optogenetic stimulation of PeFA[Ucn3] neuronal projections to the amygdalohippocampal area in a virgin female mouse during pup-directed behavior. Most females showed reduced parental behavior toward pups, and a subset of females displayed pup-directed aggression such as aggressively carrying the pups around the cage without retrieving upon light illumination as in this example.

https://elifesciences.org/articles/64680/figures#video4

the MPOA have recently been shown to modulate infant-directed behavior (*Sato et al., 2020*). We noticed that the AHi-projecting neurons that are active during infanticide were located predominantly in the rostral PeFA, as were the majority of PeFA[Ucn3] neurons associated with infanticide. Thus, we propose that the pattern of activation within subsets of PeFA[Ucn3] neurons defined by their projection targets correlates with the degree of antiparental behavior displayed by the individual, from neglect and avoidance to genuine pup-directed attack.

In turn, the roles of distinct PeFA[Ucn3] neuronal projections in the expression of pup-directed behavior may explain the different behavior outcomes we observed between somatic and projection stimulation of PeFA[Ucn3] cells. Following PeFA[Ucn3] somatic optogenetic stimulation, we observed neglect and some rare instances of aggression toward pups. Stimulation of PeFA[Ucn3] to lateral septum projections led to pup avoidance, a more pronounced phenotype than what was observed in the somatic stimulation group. Following stimulation of PeFA[Ucn3] to VMH projections, we observed pup neglect that was fairly similar to behavior observed with somatic stimulation. By contrast, stimulation of PeFA[Ucn3] to AHi projections led to overt aggressive behavior in about half the females tested, a much more pronounced response than the soma stimulation group. Thus, direct stimulation of a specific projection may be more efficient in triggering behaviors directly related to the function of the target region. For example, LS is known to have a role in persistent anxiety-like phenotypes and, indeed, stimulation of this projection elicits extreme pup avoidance. Similarly, the AHi has recently been associated with infant-directed aggression (*Sato et al., 2020*), supporting our hypothesis that this projection target is preferentially involved in pup attack behavior.

Additionally, there is a strong level of connectivity between pro-parental MPOA[Gal] and anti-parental PeFA[Ucn3]-associated circuits. Indeed, MPOA[Gal] and PeFA[Ucn] neurons have been found to project to many of the same target regions, including the VMH and LS (*Kohl et al., 2018*), and we found that about half of the monosynaptic inputs to the PeFA[Ucn3] cells originate from MPOA[Gal] neurons (*Figure 5*). Thus, the strength of MPOA[Gal] inhibitory tone in parental animal is likely to strongly affect the behavioral outcomes observed after somatic versus projection stimulation of PeFA[Ucn3] neurons.

Parental behavior, and by extension infant-directed aggression, is highly regulated by context and previous social experience. Previous studies have shown that females with repeated exposure to pups will become more parental over time (*Lonstein and De Vries, 2000*). Our experiments with virgin females also showed a similar strong effect of pup exposure on subsequent infant-directed behavior. We observed that males allowed to interact with pups during suppression of PeFA[Ucn3] neurons displayed long-lasting reduced aggression toward pups, and that activation of PeFA[Ucn3] neurons in females induced

**Figure 10.** Model of the organization and function of PeFA[Ucn3] neurons in infant-mediated neglect and aggression. PeFA[Ucn3] neurons receive input from brain areas involved in processing social cues and physiological state, stress, as well as parenting control. In turn, PeFA[Ucn3] neurons send bifurcating projections to major targets such as the amygdalohippocampal area (AHi), lateral septum (LS), and ventromedial hypothalamus (VMH). Stimulation of these projections specifically impacts various aspects and degrees of pup-directed agonistic behavior from avoidance (VMH), repulsion (LS), to aggression (AHi).

prolonged reduction in parenting. These results stand in contrast to the effects of stimulation of the MPOA[Gal] neurons, which confers rapid, reversible facilitation of parental behavior (*Wu et al., 2014*; *Kohl et al., 2018*). Together, these data suggest that PeFA[Ucn3] neurons may undergo significant plasticity, leading to long-lasting functional changes and may induce enduring changes in downstream parental circuitry.

The functional characterization of the PeFA[Ucn3] neurons and their major projection targets enabled us to uncover how this neural population plays a role in regulating pup-directed neglect and aggression, thus revealing a pathway that is distinct from the brain control of adult-directed aggressive responses. This may have implications for the understanding of infant-directed behavior in many taxa as hypothalamic structures and peptides are highly conserved across species. We found that specific projections control aspects of antiparental behavior such as the duration of pup investigation, repulsion, as well as stereotyped displays of pup-directed agonistic behavior. The discovery of a dedicated circuit node for pup-directed aggression further supports the notion that this behavior might have adaptive advantages in the wild. This study reveals a novel neural substrate for understanding how parental behavior is modulated at a circuit level and opens new avenues for studying the context- and physiological state-dependent expression of parenting in health and disease.

# Materials and methods

**Key resources table**

| Reagent type (species) or resource | Designation | Source or reference | Identifiers | Additional information |
|---|---|---|---|---|
| Strain, strain background (mouse, male and female, crossed to C57) | Ucn3::Cre | MMRRC | Mmcd 032078-UCD | Resuscitated in lab from frozen sperm |
| Recombinant DNA reagent (AAV2/1) | AAV1-EF1a-DIO-eNpHR3.0-EYFP | Addgene | RRID:Addgene_26966 -AAV1 | |
| Recombinant DNA reagent (AAV2/1) | AAV1-EF1a-DIO-EYFP | Addgene | RRID:Addgene_27056 -AAV1 | |
| Recombinant DNA reagent (AAV2/1) | AAV1-EF1a-DIO-hChR2(H134R) | | | *Wu et al., 2014* (gift of Karl Deisseroth) |
| Recombinant DNA reagent (AAV2/1) | AAV1-dio-hM3Dq-mCherry | | | *Kohl et al., 2018* |
| Recombinant DNA reagent (AAV2/1) | AAV1-FLEx-RG | | | *Kohl et al., 2018* |
| Recombinant DNA reagent (AAV2/1) | AAV1-FLEx-TVA-mCherry | | | *Kohl et al., 2018* |
| Recombinant DNA reagent (AAV2/1) | AAV1-FLEx-synaptophysin-eGFP | | | *Kohl et al., 2018* |
| Recombinant DNA reagent (AAV2/1) | AAV1-FLEx-tdTomato | | | *Kohl et al., 2018* |
| Recombinant DNA reagent (EnvA g-deleted rabies) | EnvA deltaG Rabies-eGFP | Salk Vector Core | | |
| Antibody | Anti-C-Fos (rabbit polyclonal) | Synaptic Systems | | (1:2000) |
| Antibody | Anti-oxytocin (rabbit polyclonal) | Immunostar | | (1:1000) |
| Antibody | Anti-vasopressin (rabbit polyclonal) | Immunostar | | (1:1000) |
| Antibody | Anti-GFP (rabbit polyclonal) | Novus Biologicals | | (1:2000) |
| Antibody | Anti-phospho-S6 (rabbit polyclonal) | Life Technologies | Cat# 44923G | (4 µg/IP) |
| Other | CTB-A488 | Molecular Probes | Cat# C34775 | (0.5% wt/vol) |

*Continued on next page*

*Continued*

| Reagent type (species) or resource | Designation | Source or reference | Identifiers | Additional information |
|---|---|---|---|---|
| Other | CTB-A647 | Molecular Probes | Cat# C34778 | (0.5% wt/vol) |
| Software, algorithm | Ethovision XT 8.0 | Noldus Technology | | |
| Software, algorithm | Observer XT 11 | Noldus Technology | | |
| Software, algorithm | BRAIM | | | https://gitlab.com/dulaclab/ucn3_neuron_microarray/-/tree/master/braimSourceCode/braim.R |
| Software, algorithm | Microarray analysis | | | https://gitlab.com/dulaclab/ucn3_neuron_microarray |

## Animals

Mice were maintained on a 12 hr:12 hr dark-light cycle with access to food and water ad libitum. All experiments were performed in accordance with NIH guidelines and approved by the Harvard University Institutional Animal Care and Use Committee (IACUC).

C57BL/6J mice as well as mice on a mixed C57BL/6J × 129/Sv background were used for the molecular identification of activated PeFA neurons.

The *Ucn3*::Cre BAC transgenic line (STOCK Tg(*Ucn3*-cre) KF43Gsat/Mmcd 032078-UCD) was resuscitated from sperm provided by MMRRC. This line was generated by inserting a Cre-recombinase cassette followed by a polyadenylation sequence into a BAC clone at the initiating ATG codon of the first coding exon of the *Ucn3* gene. Sperm from the mixed B6/FVB male was introduced into oocytes from a B6/FVB female, and the founding Cre+ animals were backcrossed to C57BL/6J . Mice from the F1 and F2 generations were used in experiments.

## Behavior assays

Mice were individually housed for at least 1 week prior to testing. Experiments were conducted during the dark phase under dim red light. Tests were recorded by Geovision surveillance system or by Microsoft LifeCam HD-5000, and behaviors were scored by an observed blind to experimental condition using Observer XT11 Software or Ethovision XT 8.0 (Noldus Information Technology). Animals were tested for a single behavior per session with at least 24 hr between sessions.

### Parental behavior

Parental behavior tests were conducted in the mouse's home cage as previously described (*Wu et al., 2014*; *Kohl et al., 2018*). Mice were habituated to the testing environment for 10 min. 1–2 C57BL6/J pups 1–4 days old were presented in the cage in the opposite corner to the nest. Test sessions began when the mouse first closely investigated a pup (touched the pup with its snout) and lasted for 5–15 min. In the event that the mouse became aggressive by biting and wounding the pup, the session was immediately halted and the pup was euthanized. The following behaviors were quantified: latency to retrieve, sniffing (close contact), grooming (handling with forepaws and licking), nest building, time spent in the nest, crouching, latency to attack (latency to bite and wound), aggression (roughly handling, aggressively grooming, aggressive carrying with no retrieval), and tail rattling. A 'parenting behavior' index was calculated as the sum of duration of sniffing, grooming, nest building, time spent in the nest, and crouching. Mice were categorized as 'parental' if they showed pup retrieval behavior and 'non-parental' if they did not retrieve pups. Mice were categorized as 'aggressive' if they showed bite attacks, tail rattling, aggressive pup carrying, or aggressive pup-grooming, and as 'non-aggressive' if they did not display any of these behaviors. Optogenetic experiments were limited to 5 min as previously performed for infanticidal testing, thus limiting the ability to score male-retrieving behavior that is typically displayed between 4 and 7 min (*Wu et al., 2014*; *Kohl et al., 2018*).

## Conspecific aggression

Conspecific aggression assays were performed as previously described (*Isogai et al., 2011*). Briefly, castrated males were swabbed with ~40 µL of urine from an intact male and introduced into the home cage of the test mouse for 5 min (session started at first close contact).

## Locomotor behavior

Locomotor behavior assays were performed in a 36 × 25 cm empty cage. Velocity and distance moved were tracked for 5 min.

## Feeding behavior

Feeding behavior was assessed by introducing a small piece of palatable food (chocolate chip cookie) into the far corner of a mouse's home cage. Session started when mouse approached the food and behavior was recorded for 5 min. Parameters rated were duration sniffing and eating the cookie.

## Fluorescence in situ hybridization

Fluorescence in situ hybridization (FISH) was performed as previously described (*Isogai et al., 2011*; *Wu et al., 2014*). Briefly, fresh brain tissue was collected from animals housed in their home cage or 35 min after the start of the behavior tests for IEG (Fos) studies. Only animals that engaged in a particular behavior were used. Brains were embedded in OCT (Tissue-Tek) and frozen with dry ice. 20 µm cryosections were used for mRNA in situ. Adjacent sections from each brain were collected over replicate slides to stain with multiple probes. The staining procedure is as previously described (*Isogai et al., 2011*; *Wu et al., 2014*). Complementary DNA of *Fos, ucn3, gal, crf, trh, pdyn, cart, sst, oxt, penk, avp,* and *yfp* were cloned in approximately 800–1200 base pairs (when possible) segments into pCRII-TOPO vector (Invitrogen). Antisense complementary RNA (cRNA) probes were synthesized with T7 or Sp6 (Promega) and labeled with digoxigenin (DIG; Roche) or fluorescein (FITC, Roche). Where necessary, a cocktail of two probes was generated covering different segments of the target sequence to increase signal-to-noise ratio.

Probe hybridization was performed with 0.5–1.0 ng/L cRNA probes in an oven at 68 °C. Probes were detected by horseradish peroxidase (POD)-conjugated antibodies (anti-FITC-POD at 1/250 dilution, Roche; anti-DIG-POD at 1/500 dilution, Roche). Signals were amplified by biotin-conjugated tyramide (PerkinElmer) and visualized with Alexa Fluor-488-conjugated streptavidin or Alexa Fluor 568-conjugated streptavidin (Invitrogen), or directly visualized with TSA plus cyanine 3 system, TSA plus cyanine 5 system, or TSA plus Fluorescein system (PerkinElmer). Slides were mounted with Vectashield with 4',6-diamidino-2-phenylindole (DAPI, Vector Labs).

## Laser-capture microscopy and microarray analysis

Males were sacrificed 35 min after onset of attacking behavior for peak *Fos* expression, and fresh brain tissue was frozen in OCT. 20 µm sections with alignment holes punched into the tissue by a glass pipet were prepared in a series with every other section aimed for in situ hybridization and collected on superfrost plus slides and the other for laser-capture microdissection and collected on PEN-coated membrane slides (Leica). Double FISH for *Fos* and *oxytocin* was performed as previously described (*Isogai et al., 2011*; *Wu et al., 2014*). Sections prepared for laser capture were dehydrated and stained in a 50% ethanol solution containing cresyl violet. Using a Zeiss PALM microscope, slides from adjacent FISH and laser-capture sections are aligned via tissue pinches using PALM Software (Zeiss). After alignment, tissue from the area containing *Fos+*, oxytocin+ cells, and an outside region were identified from FISH sections and collected from fresh laser sections by laser capture. For each region, tissue from six brains were pooled to prepare total RNA (RNeasy Micro Kit, Qiagen) for reverse transcription and amplification to cDNA (Ovation Pico WTA kit, Nugen). Three replicates of cDNA (n = 18 total brain samples; six pooled per replicate) were prepared for hybridization to the Affymetrix Exon ST 1.0 array as described by the manufacturer. The mouse Affymetrix Exon 1.0 ST array was preprocessed by Bioconductor packages oligo (*Carvalho and Irizarry, 2010*) and limma (*Ritchie et al., 2015*). Briefly, the raw CEL files were read into R by read.celfiles function and normalized by rma algorithm (*Irizarry et al., 2003*). Exon-level expressions were summarized into transcript level and the probes were annotated by the moex10sttranscriptcluster.db bioconductor package (MacDonald, JW: moex10sttranscriptcluster.db: Affymetrix moex10 annotation data [chip moex10sttranscriptcluster],

R package version 8.7.0). Differential gene expression was carried out by limma using limFit, eBayes, and Toptable functions (*Phipson et al., 2016*). Volcano plot was made by R from the toptable and using a p-value of 0.001 and absolute logFC ≥ 2 as a cutoff. Data are available on GEO (accession: GSE161507), and the analysis code can be found at https://gitlab.com/dulaclab/ucn3_neuron_ microarray; *Autry, 2021*; copy archived at swh:1:rev:586576e5498f53ceec0eca6c15b17c20397f804c.

## pS6 pulldowns and RNAseq analysis

Adult male mice were habituated in their home cages to a testing room for 4 hr. Males were then either presented with a 1–4 -day-old C57BL/6J pup or no stimulus and allowed to interact for 10 min, after which time behavior was recorded and the pup was removed. 60 min after stimulus presentation, males were sacrificed and a punch (1 mm diameter) was collected from a 2 -mm-thick brain slice containing the PeFA, the paraventricular nucleus, and part of the anterior hypothalamus. The tissue was processed for pS6 immunoprecipitation as described previously (*Knight et al., 2012*). Briefly, punches were held in ice-cold homogenization buffer (10 mM HEPES, 150 mM KCl, 5 mM MgCl$_2$ and containing 1X phosphatase inhibitor cocktail, 1X calyculin, protease inhibitor cocktail [EDTA free], 250 µg/mL cycloheximide, and 2.5 µL/mL Rnasin) and 10 punches were pooled per sample, with three biological replicates taken from infanticidal and naïve groups. Tissue was homogenized, centrifuged for 10 min at 2000 g, subjected to treatment with 0.7% NP40 and 20 mM DHPC, and centrifuged for 10 min at (20,000 rcf) at 4 °C to obtain the supernatant containing the input fraction. A 10 µL sample of the input was kept for sequencing and the remaining input fraction was incubated with Protein A Dynabeads (Invitrogen) conjugated to pS6 antibody (Life Technologies 44923G ) at a concentration of around 4 µg/IP for 10 min at 4 °C. Beads were collected on a magnet and washed in 0.15 M KCl wash buffer, then associated RNAs were lysed into RLT buffer and RNA was isolated using the RNeasy Micro Kit (Qiagen) according to the manufacturer's instructions. RNA was checked for quality and quantity on an Agilent tape station. High-quality samples from matched input and IP fractions were prepared using the Low-input Library Prep kit (Clontech).

Sequenced reads were mapped with STAR aligner (*Dobin et al., 2013*), version 2.5.3a, to the mm10 reference mouse genome and the Gencode vM12 primary assembly annotation (*Mudge and Harrow, 2015*), to which non-redundant UCSC transcripts were added, using the two-round mapping approach. This means that following a first mapping round of each library, the splice-junction coordinates reported by STAR, across all libraries, are fed as input to the second round of mapping. The parameters used in both mapping rounds are: outSAMprimaryFlag = "AllBest-Score", outFilterMultimapNmax="10", outFilterMismatchNoverLmax="0.05", outFilterIntron-Motifs="RemoveNoncanonical". Following read mapping, transcript and gene expression levels were estimated using MMSEQ (*Turro et al., 2011*). Following that, transcripts and genes that cannot be distinguished according to the read data were collapsed using the mmcollapse utility of MMDIFF (*Turro et al., 2014*), and the fragments per killobase million (FPKM) expression units were converted to natural logarithm of transcript per million (TPM) units since the latter were shown to be less biased and more interpretable (*Wagner et al., 2012*). In order to test for differential gene activation between the infanticidal males and naïve males, we used a Bayesian regression model (*Perez et al., 2015*). We defined the response (for each transcript or gene) as the difference between means of the posterior TPMs of the IP and input samples (i.e., paired according to the tissue punches from which they were obtained). The uncertainty in the response was computed as square root of the sum of the variances of the posterior TPMs of the IP and input samples divided by the number of posterior samples (1000). The categorical factor for which we estimated the effect was defined as the behavior of the animal, meaning infanticidal and naïve, where naïve was defined as baseline. Hence, our Bayesian regression model estimates the posterior probability of the natural logarithm of the ratio between infanticidal IP TPMs divided by input TPMs and naïve IP TPMs divided by input TPMs being different from zero. In order to define a cutoff of this posterior probability, above which we consider the activation (i.e., effect size) significant, we searched for a right mode in the distributions of posterior probabilities across all transcripts and genes (separately for each), and placed the cutoff at the local minima that separates the right mode from the left mode, which represents transcripts and genes for which the likelihood of differential gene activation is very weak and hence their posterior probability is dominated by the prior (*Figure 1— figure supplement 1A*). Data are available at GEO (accession: GSE161552), and analysis code

is available on GitHub (https://gitlab.com/dulaclab/ucn3_neuron_microarray/-/tree/master/braim-SourceCode/braim.R).

## Optogenetics

*Ucn3*::Cre mice 8–12 weeks old were used for these experiments. Pilot experiments and our previous observations indicated that mated parents were insensitive to manipulations aimed at reducing parental behaviors, so we used virgin males and females in order to maximize our opportunity to see decrements in parenting. Virgin males were prescreened for infanticidal behavior. We injected 200 nL of AAV1/DIO-hChR2-eYFP for activation or AAV1/FLEx-NpHR3.0-eYFP (RRID:Addgene_26966_AAV1) for inhibition (or AAV1/FLEx-YFP RRID:Addgene_27056_AAV1 or AAV1/FLEx-tdTomato as control virus) bilaterally into the PeFA (AP –0.6, ML ±0.3, DV –4.2). Mice recovered for 1–2 weeks and in a second surgery a dual fiber optic cannula (300 µm, 0.22 NA, Doric Lenses) was implanted 0.5 mm above the target area and affixed to the skull using dental cement. Cannula positions were as follows: PeFA = AP –0.6 mm, ML ±0.5 mm, DV –3.7 mm; LS = AP 0.3 mm ,ML ±0.5 mm, DV –3.2 mm; VMH = AP –1.6, ML ±0.5 mm, DV –5.2; AHi = AP –2.3 mm, ML ±2.5 mm, DV –4.7 mm. Mice recovered for an additional week prior to the start of behavioral testing. In pilot studies, we found that the order of testing impacted parental behavior results, so rather than randomizing test order, we alternated laser ON and OFF conditions in the same order for both control and manipulated mice. For locomotion, feeding, and conspecific aggression tests, we randomized the order of laser ON/OFF presentation. On test days, the fiber implant was connected to an optical fiber with commutator connected to either a 473 nm laser (150 mW, Laserglow Technologies), or a 460 nm LED (50 W, Prizmatix) for ChR2 stimulation, or to a 589 nm laser (300 mW, OptoEngine) for NpHR3.0 inhibition. Behavior tests were 5 min in duration for each condition and behavior category. For parental behavior assays with ChR2, the 473 nm laser or 460 nm LED was triggered in 20 ms pulses at 20 Hz for 2 s when the mouse contacted the pup with its snout, and the order of laser sessions was ON first, OFF second, ON third. For locomotor experiments with ChR2 activation, the laser was triggered (20 ms, 20 Hz, 2 s) every 10 s to approximately equivalent duration as the time laser was on during the parental sessions, and this stimulation pattern was selected based on previous studies and pilot data (*Wu et al., 2014*; *Kohl et al., 2018*). For cookie experiments with activation, the laser was triggered when the mouse contacted the cookie. For conspecific aggression experiments with activation, the laser was triggered when the test mouse made close contact with the intruder mouse. The power exiting the fiber tip was 5 mW, which translates to ~2 mW/mm$^2$ at the target region (calculator provided by Deisseroth lab at http://www.stanford.edu/group/dlab/cgi-bin/graph/chart.php). For all behavior assays with NpHR3.0, the 589 nm laser was illuminated (20 ms, 50 Hz) throughout the duration of the 5 min session at a power of 15 mW, or ~8.4 mW/mm$^2$. For parenting sessions with inhibition, the order of laser sessions was OFF first, ON second, OFF third.

## Chemogenetics

*Ucn3*::Cre virgin female mice (or Cre-negative littermates as controls) 8–12 weeks old were used for these experiments. We injected 200 nL of AAV1/DIO-HM3Dq virus bilaterally into the PeFA (AP –0.6 mm, ± ML 0.3 mm, DV –4.2 mm). Mice recovered for 2 weeks prior to behavior testing. Cre-positive and Cre-negative females were administered 0.3 mg/kg CNO dissolved in saline intraperitoneally and habituated to the testing environment for 30 min. Females were presented two C57BL6/J pups in the corner of their homecage opposite the nest and parental behaviors were recorded for 15 min.

## Anterograde tracing

AAV2.1-FLEX-tdTomato virus (UPenn Vector Core) and AAV2.1-FLEX-synaptophysin-eGFP virus (plasmid gift of Dr. Silvia Arber; custom virus prep, UNC Vector Core) were mixed at a ratio of 1:2 and stereotaxically injected unilaterally into the PeFA (Bregma coordinates AP –0.6 mm, ML –0.3 mm, DV –4.2 mm) using a Drummond Nanoject II (180 nL to cover entire structure) into adult virgin male (n = 3) and female (n = 3) *Ucn3*-cre mice. After 2 weeks, mice were transcardially perfused with 4% paraformaldehyde. Tissue was cryoprotected (30% sucrose) and serial sections were prepared on a freezing microtome (60 µm). Every third section (covering from the prefrontal cortex through the cerebellum) was immunostained using Anti-GFP antibody (described in immunohistochemistry

methods), mounted to superfrost plus slides, coverslipped with Vectashield plus DAPI, and imaged at 10× magnification using the AxioScan (Zeiss). Synaptic densities were quantified by ImageJ software after background subtraction, and a density ratio was prepared by dividing by the density at the injection site that was normalized to 100% . Density ratio was compared across brain regions by one-way ANOVA followed by Dunnett's test for multiple comparisons and sexual dimorphisms compared using a two-tailed t-test.

### Retrograde tracing with CTB

Retrograde tracing was performed in either *Ucn3*::Cre males aged 8–12 weeks injected into the PeFA bilaterally with AAV1/FLEx-BFP allowed to recover for 10 days or into C57BL6/J males aged 8 weeks prescreened for infanticide. In *Ucn3*::Cre males, pairwise injections of 80 nL of 0.5% (wt/vol) fluorescently labeled cholera toxin B subunit (CTB-488, ThermoFisher C22841, CTB-647, ThermoFisher C34788). In C57BL6/J mice, only CTB-488 was injected. After 7 days, *Ucn3*::Cre mice were transcardially perfused and brains were postfixed overnight, cryoprotected in 30% sucrose overnight, and 60 µM sections were prepared on a sliding microtome. Sections were imaged at 10× (Axioscan, Zeiss) and the fraction of BFP+ cells double-labeled with CTB-488 and CTB-647 was quantified. In control experiments, a 1:1 mixture of CTB-488 and CTB-647 was injected into the LS (AP + 0.4 mm, ML +0.5 mm, DV –3.7 mm) and AHi (AP –2.3 mm, ML +2.5 mm, DV –4.9 mm). In C57BL6/J mice, males were habituated to a testing environment for 2 hr, then presented with one foreign C57BL6/J pup in the far corner of their homecage. When the male attacked the pup, the pup was removed from the cage and euthanized. Males were perfused 80–90 min after pup exposure and brains were postfixed overnight, cryoprotected in 30% sucrose overnight, and 60 µM sections were prepared on a sliding microtome. Tissue was stained with Fos antibody (see details in immunostaining section), imaged at 10× (Axioscan, Zeiss), and the number of Fos-positive cells labeled with CTB-488 was quantified.

### Monosynaptic input tracing

A 1:1 ratio (180 nL to cover the entire region) of AAV2.1-FLEx-TVA-mCherry (avian TVA receptor) and AAV2.1-FLEx-RG (rabies glycoprotein) was stereotaxically injected unilaterally into the PeFA (Bregma coordinates AP –0.6 mm, ML 0.3 mm, DV –4.2 mm) of adult *Ucn3*::Cre male and female mice (n = 3/sex). Two weeks later, g-deleted rabies virus (deltaG-RV-eGFP) (300 nL) was injected into the same coordinate. Seven days later, mice were transcardially perfused with 4% paraformaldehyde. Tissue was post-fixed in PFA overnight and cryoprotected (30 % sucrose), then serial sections were prepared on a freezing microtome (60 µm). Every third section (covering from the prefrontal cortex through the cerebellum) was imaged at 10× using the AxioScan (Zeiss). Cell bodies labeled by eGFP in anatomical areas were quantified and an input ratio was prepared by dividing the number of input cells in each region by the total number of input cells in each brain. Input ratio was compared across brain regions by one-way ANOVA followed by Dunnett's test for multiple comparisons and sexual dimorphisms compared using a two-tailed t-test.

### Immunohistochemistry

#### Fos antibody staining

To visualize Fos protein in combination with CTB, perfused tissue was sliced on a freezing microtome at 30 µM, and every third section throughout the PeFA was stained. Briefly, sections were rinsed in PBS with 0.1% Triton (PBST), then blocked in 2% normal horse serum and 3% fetal bovine serum diluted in PBST for 2 hr at room temperature. Primary antibody rabbit anti-Fos (Synaptic Systems) was diluted 1:2000 in PBS and sections were incubated for 72 hr at 4 °C. After rinsing with PBST, secondary anti-rabbit A568 (1:1000) was applied at 1:1000 dilution in PBS for 48 hr at 4 °C. Sections were rinsed in PBS, mounted to superfrost plus slides, coverslipped with Vectashield containing DAPI, and imaged (see details above).

#### GFP antibody staining

To amplify synaptophysin densities in the anterograde tracing experiments, the GFP signal was enhanced by antibody staining. Perfused tissue was sliced on a freezing microtome at 60 µM, and every third section throughout the brain was stained. Briefly, sections were rinsed in PBS with 0.1% Triton (PBST), then blocked in 1.5% normal horse serum diluted in PBST for 1 hr (blocking solution)

at room temperature. Primary antibody Rrabbit anti-GFP (Novus Biologicals NB600-308) was diluted 1:2000 in blocking solution and sections were incubated overnight at 4 °C. After rinsing with PBST, secondary anti-rabbit-A488 was applied at 1:200 dilution in blocking buffer and incubated overnight at 4 °C. Sections were rinsed in PBS, mounted to superfrost plus slides, coverslipped with Vectashield containing DAPI, and imaged (see details above).

## Oxytocin/vasopressin staining
Perfused tissue was sliced on a sliding microtome at 60 µM. Every third section from the PeFA was stained. Briefly, sections were rinsed in PBS with 0.1% Triton (PBST), then blocked in 10% fetal bovine serum diluted in PBST. Primary antibody rabbit anti-vasopressin (Immunostar) or rabbit anti-oxytocin (Millipore) were diluted 1:1000 in PBST and sections were incubated overnight at 4 °C. After rinsing with PBST, secondary anti-rabbit A647 was applied at 1:200 dilution in blocking buffer and incubated overnight at 4 °C. Sections were rinsed in PBS, mounted to superfrost plus slides, coverslipped with Vectashield containing DAPI, and imaged on the Axioscan (Zeiss) at 10× magnification.

## Electrophysiology
### Solutions
We used modified artificial cerebrospinal fluid (ACSF) contained (in mM): 105 choline chloride, 20 glucose, 24 $NaHCO_3$, 2.5 KCl, 0.5 $CaCl_2$, 8 $MgSO_4$, 5 sodium ascorbate, 3 sodium pyruvate, 1.25 $NaH_2PO_4$ (osmolarity 290, pH 7.35) for dissecting out the brain. All recordings were made in an oxygenated ACSF with composition (in mM): 115 NaCl, 2.5 KCl, 25 $NaHCO_3$, 1.25 $NaH_2PO_4$, 1 $MgSO_4$, 20 glucose, 2.0 $CaCl_2$ (osmolarity 290, pH 7.35).

Voltage-clamp internal solution contained (in mM): 130 d-gluconic acid, 130 cesium hydroxide, 5.0 NaCl, 10 HEPES, 12 di-tris-phosphocreatine, 1 EGTA, 3.0 Mg-ATP, and 0.2 $Na_3$-GTP (osmolarity 291, pH 7.35). All chemicals were purchased from Sigma-Aldrich.

### Acute brain slices
8–12 -week-old *Ucn3*::Cre male (n = 5) and female (n = 3) mice were stereotaxically injected with AAV1-eF1a-DoubleFlox hChR2(H134R)-mCherry-WPRE-HgHpA (Addgene, Cambridge, MA) bilaterally into the PeFA (AP –0.6, ML ±0.3, DV –4.2; 200 nL). Mice recovered from surgery for 2–3 weeks and then sacrificed. Coronal brain slices were prepared using methods described previously (*Kapoor and Urban, 2006*). Mice were lightly anesthetized with isoflurane exposure using a vaporizer (Datex-Ohmeda) connected to a clear acrylic chamber for 2 min, and then deeply anesthetized with an i.p. injection of a mixture of ketamine (100 mg/kg) and xylazine (10 mg/kg). Ice-cold modified ACSF cutting solution was used to transcardially perfuse the mice, and brains were dissected into the same solution. Coronal slices (300 µm thick) were obtained using a vibratome (VT1000S; Leica, Germany) and collected in ice-cold cutting solution. After cutting, slices were incubated in oxygenated ACSF solution at 35 °C for 45 min and then at room temperature for the duration of the experiment.

## In vitro recordings
Whole-cell voltage-clamp recordings were made using borosilicate glass patch pipettes (6–10 MΩ) filled with internal solutions, respectively, and slices maintained at 35 °C in oxygenated ACSF. Slices were visualized under custom-built infrared optics on a BX51WI microscope (Olympus Optical, Tokyo, Japan). Recordings were obtained with a Multiclamp 700B amplifier (Molecular Devices, Palo Alto, CA), and physiological data were collected via software written in LabView (National Instruments) and pClamp 10.3 (Molecular Devices).

For optogenetic photostimulation experiments, ChR2 was activated using custom-built 465 nm (1–5 mW/mm², CBT-90-B-L11, Luminus) light source. Light pulses of 5–20 ms, delivered at 2–20 Hz, were used to probe synaptic connectivity. We used Wilcoxon rank-sum test to compare the distributions.

For chemogenetics stimulation experiments, 8–12- week-old *Ucn3*::Cre male mice were stereotaxically injected with pAAV-hSyn-DIO-hM3D(Gq)-mCherry (Addgene) bilaterally into the PeFA (AP –0.6, ML ±0.3, DV –4.2; 200 nL). For the stock solutions, CNO was dissolved at 0.5 mg/mL concentrations in ACSF and was further diluted to 10 µM in ACSF for the experiments. We measured the effects of CNO on *Ucn3*::Cre*hChR2(H134R)-mCherry neurons by selectively recording from mCherry-positive neurons in PeFA. The recording protocol included 3–5 min of baseline followed by 3–5 min of bath

application of 10 µM CNO and 3–5 min of washout of CNO. We used Wilcoxon rank-sum test to compare the distributions of firing rate.

## Statistics

In our molecular studies, we used one-way ANOVA followed by Tukey's post-hoc test or Kruskal–Wallis test followed by Dunn's post-hoc test. Details for differential gene expression for the microarray analysis are explained under 'Microarray' in the Materials and methods section. Similarly, the pS6 data analysis is described in the 'pS6 pulldowns and RNAseq analysis' section.

For most optogenetics behavior studies, we used repeated one- or two-way ANOVA analyses followed by Bonferroni or Tukey's post-hoc tests. For the pup retrieval percentage data, we used a Friedman test. To compare numbers of animals retrieving across two trials, we used a Fisher's exact test or chi-square test, and across three trials, we used an accelerated failure regression model. For DREADD experiments, we used t-test or Mann–Whitney test.

For the projection and input data, we used regression analysis.

For electrophysiology analysis, we used two-sided Wilcoxon rank-sum test to compare the amplitudes of evoked EPSCs across regions (LS, VMH, and AHi) and for comparing the amplitudes of EPSCs in paired pulse paradigm.

## Replication and outliers

For molecular experiments, we used 3–6 biological replicates (mice) per group. For animal behavior experiments, we used 9–12 animals per group based on our previous experience with studying parental behavior. For all experiments, both molecular and behavioral, experiments were performed in at least two cohorts if not more.

For molecular experiments, outliers were determined using a Grubb's outlier test. For behavioral experiments, mice were excluded if post-hoc analysis revealed insufficient viral infection.

## Acknowledgements

We thank S Sullivan for help with behavior scoring and mouse husbandry, and R Hellmiss at MCB Graphics for work on illustrations. We thank members of the Dulac lab for comments on the manuscript, A Leonard, E Owolo, and L Moussalime for assistance with data collection. This work was supported by a NIH K99 and R00 Award (K99HD085188, R00HD085188) and a NARSAD Young Investigator Award to AEA, a Human Frontier Long-Term Fellowship, a European Molecular Biology Organization Long-Term Fellowship (ALTF 1008-2014) and a Sir Henry Wellcome Fellowship from the Wellcome Trust to JK, a NIH K99 Award (K99HD092542) to DJBM, a Howard Hughes Gilliam Fellowship for Advanced Study to BMB, and NIH grant 1R01HD082131-01A1 to CD. CD is an investigator of the Howard Hughes Medical Institute.

## Additional information

### Competing interests

Catherine Dulac: Senior editor, *eLife*. The other authors declare that no competing interests exist.

### Funding

| Funder | Grant reference number | Author |
| --- | --- | --- |
| Eunice Kennedy Shriver National Institute of Child Health and Human Development | K99HD085188 | Anita E Autry |
| Brain and Behavior Research Foundation | NARSAD Young Investigator | Anita E Autry |
| European Molecular Biology Laboratory | ALTF 1008-2014 | Johannes Kohl |

| Funder | Grant reference number | Author |
| --- | --- | --- |
| Wellcome Trust | Sir Henry Wellcome Fellowship | Johannes Kohl |
| National Institute of Mental Health | K99HD092542 | Dhananjay Bambah-Mukku |
| Eunice Kennedy Shriver National Institute of Child Health and Human Development | 1R01HD082131-01A1 | Catherine Dulac |
| Howard Hughes Medical Institute | HHMI investigator | Catherine Dulac |

The funders had no role in study design, data collection and interpretation, or the decision to submit the work for publication.

## Author contributions

Anita E Autry, Conceptualization, Data curation, Formal analysis, Funding acquisition, Investigation, Methodology, Validation, Visualization, Writing - original draft, Writing – review and editing; Zheng Wu, Conceptualization, Data curation, Formal analysis, Investigation, Methodology, Writing – review and editing; Vikrant Kapoor, Conceptualization, Data curation, Formal analysis, Investigation, Methodology, Validation, Visualization, Writing - original draft, Writing – review and editing; Johannes Kohl, Data curation, Formal analysis, Investigation, Methodology, Writing – review and editing; Dhananjay Bambah-Mukku, Data curation, Formal analysis, Investigation, Methodology, Validation, Writing – review and editing; Nimrod D Rubinstein, Data curation, Formal analysis, Investigation, Methodology, Software; Brenda Marin-Rodriguez, Ilaria Carta, Victoria Sedwick, Data curation, Formal analysis; Ming Tang, Formal analysis, Software; Catherine Dulac, Conceptualization, Formal analysis, Funding acquisition, Investigation, Methodology, Project administration, Supervision, Visualization, Writing – review and editing

## Author ORCIDs

Anita E Autry (ID) http://orcid.org/0000-0003-3933-1977
Vikrant Kapoor (ID) http://orcid.org/0000-0002-0185-3836
Johannes Kohl (ID) http://orcid.org/0000-0001-8222-0282
Ilaria Carta (ID) http://orcid.org/0000-0003-1270-573X
Catherine Dulac (ID) http://orcid.org/0000-0001-5024-5418

## Ethics

All animal experiments were approved by the Harvard University Institutional Animal Care and Use Committee. All experiments were performed in compliance with our Harvard University IACUC approved protocols 97-03-3, 23-12-3, and 25-13-3.

## Decision letter and Author response

Decision letter https://doi.org/10.7554/eLife.64680.sa1
Author response https://doi.org/10.7554/eLife.64680.sa2

# Additional files

## Supplementary files

• Transparent reporting form

## Data availability

Microarray data have been deposited in GEO under accession code GSE161507 and analysis code can be found at https://gitlab.com/dulaclab/ucn3_neuron_microarray copy archived at https://archive.softwareheritage.org/swh:1:rev:586576e5498f53ceec0eca6c15b17c20397f804c. pS6 data have been deposited in GEO under accession code GSE161552 and analysis code is available on Github (https://gitlab.com/dulaclab/ucn3_neuron_microarray/-/tree/master/braimSourceCode/braim.R).

The following dataset was generated:

| Author(s) | Year | Dataset title | Dataset URL | Database and Identifier |
|-----------|------|---------------|-------------|-------------------------|
| Autry AE, Tang M | 2021 | Expression data from adult male mouse hypothalamus | https://www.ncbi.nlm.nih.gov/geo/query/acc.cgi?acc=GSE161507 | NCBI Gene Expression Omnibus, GSE161507 |
| Autry AE, Tang M | 2021 | Expression data from adult male mouse hypothalamus | https://www.ncbi.nlm.nih.gov/geo/query/acc.cgi?acc=GSE161552 | NCBI Gene Expression Omnibus, GSE161552 |

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
