## [Decision Letter]

**Acceptance summary:**

While previous studies identify the brain biology mediating positive parenting toward infants, this impactful study provides new information about the brain mechanisms mediating negative behaviors toward infants. Using an elegant and multi-level approach, this study uncovers brain cell subtype and circuit underpinnings that regulate infant directed aggression and neglect. Collectively, this paper provides a new framework to understand the positive and negative regulation of parenting behavior.

**Decision letter after peer review:**

Thank you for submitting your article "Urocortin-3 Neurons in the Mouse Perifornical Area Promote Infant-directed Neglect and Aggression" for consideration by *eLife*. Your article has been reviewed by 3 peer reviewers, including Mary Kay Lobo as Reviewer #1 and the Reviewing Editor, and the evaluation has been overseen by Kate Wassum as the Senior Editor. The following individuals involved in review of your submission have agreed to reveal their identity: Christian Broberger (Reviewer #2); Lauren A O'Connell (Reviewer #3).

The reviewers have discussed the reviews with one another and the Reviewing Editor has drafted this decision to help you prepare a revised submission.

Summary:

This study makes an important contribution to our understanding of the brain mechanisms that drive instinctive behaviors, by identifying a neuronal population involved in pup-directed aggression. By applying a broad array of techniques this study provides compelling evidence that urocortin III (Ucn3) neurons in the PeFA induce infanticide behavior in male mice. While previous studies identify circuits mediating positive parenting toward infants this impactful study provides new information into the circuitry mediating negative behaviors toward infants. This paper will be of interest to scientists studying both parental and/or aggression behavior and the underlying neural circuits.

Essential revisions:

There is some difficulty synthesizing these data into a unified model of the exact role of these neurons in adult mouse responses to pups, these limitations can be amended through mainly text edits. Additional text edits, figure edits, inclusion or quantification of new images, and clarification or inclusion of new electrophysiology data will further help to improve clarity and support the conclusions in this manuscript. Detailed comments are listed below.

1. In the introduction, the framing is centered on mammals, but this work rests on a broader framework of behavioral ecology, where many taxa (in addition to mammals) invest considerable resources in offspring. Likewise, infanticide is widespread across the animal kingdom (birds, fish, frogs, invertebrates). It is encouraged that the authors highlight some non-mammalian species to expand the behavioral ecology foundation on which this work could stand and make it of interest to a wider audience. This could be applied in the first paragraphs of the introduction and discussion.

2. Edits to the text would provide more clarity to support the conclusions about the neural basis of pup directed aggression.

a. The data as they stand appear to only support a role for PeFAUcn3 cells driving pup directed aggression in males, while in females activity of the neurons disrupt pup care but does not unambiguously drive aggression. Please rephrase to reflect in which instances – if any – (e.g. anatomy, activating stimuli, effects of stimulating or inhibiting) the authors believe the cells to be different between the sexes and what this might mean functionally. Or do the authors think that the findings reflect the fact that in male inhibition paradigms were primarily employed, whereas females were primarily exposed to activation?

b. There seems to be a lot of variation in behavior and even a bimodal response sometimes. For example, when PeFAUcn3 neuron are inhibited, only three males have inhibited pup-directed aggression, but all lack aggression by the next OFF period. Please discuss as to why there is this variation in infant-directed aggression in males and how do the authors posit this is regulated within the present circuit. Additional discussion into why these OFF manipulations do not immediately work in all males is warranted. Is there a circuit activity basis for the time delay?

c. It would be very valuable if the authors comment in the Discussion about why the combined results of stimulating the three PeFAUcn3 projections do not seem to mirror the effect of stimulating the cell bodies. It is not expected to see the "full" effect of somatic illumination (as additional, non-tested projections may contribute to that), but there appears to be more aggression with terminal than with cell body activation. It would benefit the reader to have the authors' provide discussion to make sense of this discrepancy.

d. Only 2-3 animals out of 15 females showed signs of aggression thus describing that "activation of PeFAUcn3 neurons…induced signs of aggression" at the end of the first paragraph on page 8 may be an overstatement.

e. In the discussion, the authors state that PeFAUcn3 neuron activation is "long-lasting and not reversible." Please revise this as "Not reversible" is an overstatement given the authors examined this effect only 24 hours later, but not at a longer time period.

f. Did the authors manipulate the activity of PeFAUcn3 neurons in parents (dams, sires) and if so, what was the effect? It is not necessary to perform this study, but if is available it would provide further knowledge into the role of PeFA Ucn3 neurons in behavior.

3. Additional text edits would provide clarity to this the study and support the conclusions made by the authors.

a. In Figure 1H the authors show data from virgin females that engaged in pup attacks but Figure 3G and Figure 4E show control females do not display aggressive behavior. Please provide more information into how female pup attacks were quantified in Figure 1 vs Figure 3 and 4 that would account for these differences.

b. In the first paragraph for the expression analyses of c-fos positive brain areas using in situ hybridization and subsequent laser microdissection, did all virgin males attack? The text only mentions "sections of tissue from virgin males after interactions with pups" – it does not hint at the outcome of the behavioral interaction.

c. For the text corresponding to figure 1E – "In situ hybridization analysis showed that Ucn3 expression is spatially restricted to the PeFA…" Please clarify what this is compared to and is this the only place in the brain that this peptide is made?

d. In the Results, when data is reported from "infanticidal males", were these animals certified infanticidal, i.e. had they been tested beforehand in the presence of pups?

e. In the discussion, the authors state "In our experiments, MPOAGal neurons remain fully activated in females exposed to pups, thus exerting their inhibitory role on PeFAUcn3 neurons and their projections". Please clarify where the authors show that MPOA Gal neurons exert inhibitory role on PeFAUnc3 neurons. Based on Figure 5E, it does not look like many of these Gal neurons project to the Ucn3 neurons and there is no information about activity-based regulation between the two. However, Figure 5F demonstrates high colocalization. Can the authors please clarify the connection between these two cell types – both based on the evidence presented here and their hypotheses about how these cells may regulate one another? If there is no demonstration of an inhibitory role then the authors may rephrase to clarify that this is inference from the GABAergic molecular phenotype of MPOAGal neurons since GABA transmission can in some cases be depolarizing.

f. The authors state that Ucn3 neurons in the PeFA are involved in the "regulation of social recognition" and then go on to say that these neurons are tuned specifically to cues from pups. Please clarify if adult social recognition was also examined?

4. Additional clarity about statistical methods and data analyses employed.

a. The Methods lack a statistics section. There does not appear to be a description of statistics for behavior data, whether the many comparisons met the assumptions of the many parametric tests, or what software was used for statistics in many of the comparisons. The reviewers do acknowledge that some statistical methods were presented, such as with gene expression data.

b. Code for Bayesian model of pS6 data could not be located on the indicated GitHub page provided in the manuscript. Either it is not there currently, or it is not well labeled. Please make sure this code is clearly labeled, well annotated, and available prior to publication.

c. In reference to the sex differences uncovered in Figure 1 – Colocalization of c-fos and Ucn3 was significant only in virgin males that attack pups and not females although a trend was observed. Did the authors do a power analysis prior to this study? Is this a power issue or is there a real sex difference here?

d. Figure 1 – Supplement 1 Graph E. There are t-test results but please clarify that a t-test was performed in the legend? It appears that a t-test is comparing the colocalization to total Ucn3+ neurons, but if that is the case, an independent t-test is not appropriate.

e. In Figure 7 use caution when using a t-test with only 3 replicates and make sure these data fit the assumptions of this parametric test.

f. Most of the Figure legends include statistical information but Figure 8 legend is missing n and statistics.

g. please ensure each statistic has full reporting, e.g., test statistic, degrees of freedom, in addition to p value.

5. New images or quantification of images will improve the conclusions in this manuscript.

a. Figure 1B is the main figure that demonstrates visualization of enhanced c-Fos reactivity in PeFA of virgin males after infant directed attacks. However, the low power image used does not easily convey this information. While the authors include many higher power images in other subfigures it would be helpful for the reader to see a higher power image inlet in this first subfigure.

b. On page 6 the authors state that excitatory Ucn3 neurons are activated based on the Vglut/Ucn3 colocalization image in Supplemental Figure 1. However, it is difficult to visualize the colocalization in this image. Please provide quantification of the Vglut/Ucn3 positive neurons relative to Ucn3 only positive neurons.

c. The images shown in Supplemental Figures 2, 3 and 4 are not ideal for demonstrating that the construct is expressed in the Ucn3 cells. Please show double-staining for the virus-encoded fluorophore and Ucn3 mRNA/protein. Inclusion of quantification for at least one construct would demonstrate the level of expression in Ucn3 neurons.

6. Edits to the electrophysiology data in Figure 8 and validation of optogenetic/ chemogenetic protocols.

a. Please justify the relevance of including the data in Figure 8A-C as pretty much all brain neurons are expected to receive excitatory input and this data set does not isolate PeFA synapses.

b. Using amplitude as a measure of the strength of connectivity appears to be an oversimplification as it is difficult to make a direct comparison between targets using the experimental design in this experiment.

c. The authors classify the PSCs they recorded after stimulation of PeFAUcn3 cells as excitatory. This is a fair assumption given that these cells are glutamatergic but the manuscript lacks information specifying that there was pharmacological isolation by gabazine or a similar drug. Please clarify this.

d. Figure 8A-C: As noted above there is uncertainty as to the value of these data in the context of the current study. However, if they are included then please also include raw traces and not only quantification.

e. Please include validation (eg by slice electrophysiology and application of light/CNO) that show that the opto- and chemogenetic strategies for stimulation and inhibition achieves the in vivo stimulation goals at a cellular level.

---

## [Author Response]

Essential revisions:There is some difficulty synthesizing these data into a unified model of the exact role of these neurons in adult mouse responses to pups, these limitations can be amended through mainly text edits. Additional text edits, figure edits, inclusion or quantification of new images, and clarification or inclusion of new electrophysiology data will further help to improve clarity and support the conclusions in this manuscript. Detailed comments are listed below.1. In the introduction, the framing is centered on mammals, but this work rests on a broader framework of behavioral ecology, where many taxa (in addition to mammals) invest considerable resources in offspring. Likewise, infanticide is widespread across the animal kingdom (birds, fish, frogs, invertebrates). It is encouraged that the authors highlight some non-mammalian species to expand the behavioral ecology foundation on which this work could stand and make it of interest to a wider audience. This could be applied in the first paragraphs of the introduction and discussion.

Thanks for the excellent suggestion, which indeed broadens the impact of our study. We have added information about infanticide and parental switch across a wide range of animal species in both the introduction and the discussion.

2. Edits to the text would provide more clarity to support the conclusions about the neural basis of pup directed aggression.a. The data as they stand appear to only support a role for PeFAUcn3 cells driving pup directed aggression in males, while in females activity of the neurons disrupt pup care but does not unambiguously drive aggression. Please rephrase to reflect in which instances – if any – (e.g. anatomy, activating stimuli, effects of stimulating or inhibiting) the authors believe the cells to be different between the sexes and what this might mean functionally. Or do the authors think that the findings reflect the fact that in male inhibition paradigms were primarily employed, whereas females were primarily exposed to activation?

As correctly pointed out by the reviewers, we observe a robust reduction of aggression in virgin males during optogenetic inhibition of PeFA^Ucn3^ neurons, while in virgin females, activation of this neuronal population triggers neglect, rather than aggression. However, we also show that specific stimulation of the PeFA^Ucn3^ projections to AHi generates clear observable aggressive behavior of virgin females toward pups. Thus, PeFA^Ucn3^ neuron activity may trigger aggression of both males and females toward pups, as further elaborated below (see also further response to point 2c regarding the stimulation of PeFA^Ucn3^ projections).

In addition, we have also manipulated the activity of PeFA^Ucn3^ neurons in mated males and females (see also response to point 2f about activity manipulation in parents) and did not observe changes in behavior. As mentioned in later points, we have now added a sentence in the discussion to summarize and clarify the entire set of functional manipulations.

Our interpretation of the lack of observable aggression during artificial activation of PeFA^Ucn3^ neurons in animals normally displaying parenting behavior (virgin females, mothers and fathers) relies on our findings that PeFA^Ucn3^ neurons receive substantial inhibitory input from MPOA^Gal^ neurons (Figure 5 E, F). Because MPOA^Gal^ neurons are highly active in all parenting animals in response to pup cues, particularly in mated males and females (mothers and fathers) (Wu *et al.*, 2014; Kohl *et al.*, 2018), we postulate that aggression is readily observed in virgin males after artificial activation of PeFA^Ucn3^ neurons because MPOA^Gal^ neurons do not exert any inhibitory modulation of PeFA^Ucn3^ neurons in that state.

By contrast, the direct stimulation of the PeFA^Ucn3^ projections to AHi in parental animals may effectively shortcut the inhibition by MPOA^Gal^ neurons, as well as specifically focus the outcome of PeFA^Ucn3^ activity to a strong aggression-driving brain area (AHi) and thus leads to observable aggressive displays toward pups. This hypothesis is further elaborated in the response to point 2c below.

Our summary model, presented in figure 10 proposes that PeFA^Ucn3^ neurons are tuned to pup-directed neglect and aggression in both males and females, but undergo substantial inhibition of activity in parenting animals (virgin females, mothers, and fathers). To formally demonstrate this model, one would need to alleviate the inhibition from the brain circuitry normally mediating parenting, meaning inhibit MPOA^Gal^ neurons, while activating PeFA^Ucn3^ neurons, in order to robustly trigger aggression in parenting animals. Unfortunately, we do not yet have the appropriate tools to manipulate two genetically identified neural populations. We have clarified our findings and interpretation in the discussion and emphasized our view on the critical role of MPOA^Gal^ inhibitory inputs on the PeFA^Ucn3^ neural population in parenting animals.

b. There seems to be a lot of variation in behavior and even a bimodal response sometimes. For example, when PeFAUcn3 neuron are inhibited, only three males have inhibited pup-directed aggression, but all lack aggression by the next OFF period. Please discuss as to why there is this variation in infant-directed aggression in males and how do the authors posit this is regulated within the present circuit. Additional discussion into why these OFF manipulations do not immediately work in all males is warranted. Is there a circuit activity basis for the time delay?

When PeFA^Ucn3^ neuron are inhibited, we indeed observed complete suppression of aggression in 3 males (Figure 2c), and found that all but one male showed an extremely long latency to attack and a significantly reduced percentage of time spent displaying aggression (Figure 2d). Thus overall, PeFA^Ucn3^ neuron inhibition induces a clear reduction of aggression towards pups. Interestingly, this inhibition appears to be long lasting and even further progress, such that all males are non-aggressive by the time of the next behavior testing (OFF 2). At the circuit level, we interpret this prolonged and self-sustained reduction in pup-mediated attack as an indication that pro-parental circuits become progressively more active following inhibition of PeFA^Ucn3^ neurons, thus leading to a complete loss of aggression by the time we retest the males.

The apparent bimodal distribution during the ON trial may result from a distribution of efficiency of virus infection biased toward different projection targets. We quantified viral efficiency, but this parameter did not explain the distribution in behavior. We have included a statement clarifying our interpretation that suppression of PeFAUcn3 activity favors expression of parental behavior in the discussion.

c. It would be very valuable if the authors comment in the Discussion about why the combined results of stimulating the three PeFAUcn3 projections do not seem to mirror the effect of stimulating the cell bodies. It is not expected to see the "full" effect of somatic illumination (as additional, non-tested projections may contribute to that), but there appears to be more aggression with terminal than with cell body activation. It would benefit the reader to have the authors' provide discussion to make sense of this discrepancy.

Thank you for pointing out this issue: we agree that a more explicit discussion of the data obtained by stimulating PeFA^Ucn3^ somas versus projections is needed. PeFA^Ucn3^ soma stimulation induces neglect behavior and some instances of aggression toward pups in 3/15 females. Stimulation of PeFA^Ucn3^ to ventromedial hypothalamus projection triggers neglectful behavior, fairly similar to behavior observed in the somatic stimulation group. Stimulation of PeFA^Ucn3^ to lateral septum projection triggers pup avoidance, a more pronounced phenotype than in the soma stimulation group. Finally, stimulation of PeFA^Ucn3^ to AHi projection triggers aggressive behavior in about half the females tested, which as the reviewer has noted is a much more pronounced aggression than the soma stimulation group. Our interpretation involves two non-exclusive phenomena. First, inhibitory MPOA^Gal^ neurons project to Ucn3+ neuronal somas in the PeFA as well as to many of the same target regions as PeFA^Ucn3^ neurons, including the VMH and LS (Kohl *et al.*, 2018). Thus, it is possible that the strength of agonistic phenotypes observed by stimulating PeFA^Ucn3^ neuronal soma versus various projections depends on the strength of the PeFA^Ucn3^ excitatory connection to these target areas as well as that of MPOA^Gal^ inhibitory tone in parental animals. Second, direct stimulation of a specific projection may be more efficient in triggering behaviors directly related to the function of the target region. For example, LS is known to have a role in persistent anxiety-like phenotypes and indeed, stimulation of this projection elicits extreme pup avoidance. Similarly, the AHi has recently been associated with infant-directed aggression (Yamaguchi et al., 2020), supporting our hypothesis that this projection target is preferentially involved in pup attack behavior. We have amended the discussion to include these hypotheses.

d. Only 2-3 animals out of 15 females showed signs of aggression thus describing that "activation of PeFAUcn3 neurons…induced signs of aggression" at the end of the first paragraph on page 8 may be an overstatement.

Thank you for the observation: this is indeed an overstatement, and we have now amended the text to reflect that we observed signs of aggression in only a few females.

e. In the discussion, the authors state that PeFAUcn3 neuron activation is "long-lasting and not reversible." Please revise this as "Not reversible" is an overstatement given the authors examined this effect only 24 hours later, but not at a longer time period.

We agree with the reviewer that “not reversible” is too strong of a statement in view of the short 24 hour time window used in our assay. We have now amended the statement to reflect this comment (see also point 2b).

f. Did the authors manipulate the activity of PeFAUcn3 neurons in parents (dams, sires) and if so, what was the effect? It is not necessary to perform this study, but if is available it would provide further knowledge into the role of PeFA Ucn3 neurons in behavior.

As referred to in an earlier part of the response to reviewers, we did attempt to activate PeFA^Ucn3^ neurons in mothers and fathers, and we saw no effect on behavior. We have a statement in the “Optogenetics” methods section that provides details.

We believe that these data highlight the importance of physiological state in the regulation of parenting versus infanticide and hope to follow up on these data in the future using more refined approaches. Therefore, we have also added a similar statement in the main text in the Results section and in the discussion explaining the experiments in mated animals and our interpretation.

3. Additional text edits would provide clarity to this the study and support the conclusions made by the authors.a. In Figure 1H the authors show data from virgin females that engaged in pup attacks but Figure 3G and Figure 4E show control females do not display aggressive behavior. Please provide more information into how female pup attacks were quantified in Figure 1 vs Figure 3 and 4 that would account for these differences.

Thank you for pointing out this issue. We rarely saw pup-directed aggression in females, but in the process of testing many animals for the data shown in Figure 1H, we did come across a few infanticidal females that attacked pups. In subsequent experiments, we did not observe any pup directed attack, but detected some aggressive displays including tail-rattling, aggressive carrying, and biting. We have added a sentence to indicate that pup-directed aggression is rare in females, but we did observe attack in 3 mice over the course of our screen. Later in the text, we specify that the aggressive behavior quantified is tail-rattling behavior for 3G and bite-attack and aggressive carrying for 4E. We give a detailed summary of our evaluation criteria in the methods section regarding aggressive and parental behavior quantification.

b. In the first paragraph for the expression analyses of c-fos positive brain areas using in situ hybridization and subsequent laser microdissection, did all virgin males attack? The text only mentions "sections of tissue from virgin males after interactions with pups" – it does not hint at the outcome of the behavioral interaction.

Thank you for pointing out this issue. We have clarified this point by amending the text as follows: "sections of tissue from virgin males after infant-directed attack".

c. For the text corresponding to figure 1E – "In situ hybridization analysis showed that Ucn3 expression is spatially restricted to the PeFA…" Please clarify what this is compared to and is this the only place in the brain that this peptide is made?

Thank you for identifying this concern. According to the literature, Ucn3 is expressed in the PeFA but not in the adjacent paraventricular nucleus. Ucn3 is also expressed in the MPOA, the medial amygdala, the BNST/PeFA and the nucleus ambiguus (Venihaki *et al.*, 2004; Wittmann *et al.*, 2009; Deussing *et al.*, 2010). We have added these references in the text corresponding to figure 1E.

d. In the Results, when data is reported from "infanticidal males", were these animals certified infanticidal, i.e. had they been tested beforehand in the presence of pups?

Yes, all males categorized as infanticidal displayed pup-directed attack during a pup exposure test.

e. In the discussion, the authors state "In our experiments, MPOAGal neurons remain fully activated in females exposed to pups, thus exerting their inhibitory role on PeFAUcn3 neurons and their projections". Please clarify where the authors show that MPOA Gal neurons exert inhibitory role on PeFAUnc3 neurons. Based on Figure 5E, it does not look like many of these Gal neurons project to the Ucn3 neurons and there is no information about activity-based regulation between the two. However, Figure 5F demonstrates high colocalization. Can the authors please clarify the connection between these two cell types – both based on the evidence presented here and their hypotheses about how these cells may regulate one another? If there is no demonstration of an inhibitory role then the authors may rephrase to clarify that this is inference from the GABAergic molecular phenotype of MPOAGal neurons since GABA transmission can in some cases be depolarizing.

We agree that our phrasing was too strong and have clarified how previously published work, as well as our new data, have led to our current hypothesis of an inhibitory effect of MPOA^Gal^ neurons on the PeFA^Ucn3^ of parental animals. In brief, previous work has demonstrated that in females and fathers, MPOA^Gal^ neurons are highly active (Wu *et al.*, 2014; Kohl *et al.*, 2018). Moreover, we observed in the present study that around half of MPOA neurons projecting to PeFA^Ucn3^ neurons express galanin, thus representing a highly substantial input category to these neurons (Figure 5e,f). To further clarify and emphasize these data, we have updated the figure legend to indicate that the arrowheads depict colocalized cells (5e) and that the percentages are for the colocalized inputs (5f). Thus, we hypothesize that MPOA^Gal^ neurons, which are around 95% GABAergic (Wu *et al.*, 2014), inhibit the PeFA^Ucn3^ neurons in parentally-behaving mice. As the reviewer rightfully pointed out, we do not have direct evidence for inhibition of the PeFA^Ucn3^ neurons by the MPOA^Gal^ inputs, and have therefore rephrased our statement in the text to indicate that we infer that the galanin neurons are likely inhibiting the PeFA^Ucn3^ cells, as also discussed in response to Point 2a. Amended as follows: “In our experiments, MPOA^Gal^ neurons remain fully activated in females exposed to pups, thus exerting their putative inhibitory role on PeFA^Ucn3^ neurons and their projections”.

f. The authors state that Ucn3 neurons in the PeFA are involved in the "regulation of social recognition" and then go on to say that these neurons are tuned specifically to cues from pups. Please clarify if adult social recognition was also examined?

Previous literature has examined the role of PeFA^Ucn3^ neurons in the juvenile social recognition test (Deussing *et al.*, 2010). We did not test juvenile recognition in the present study, but tested social interaction with an adult intruder (castrated male swabbed with male urine) in data provided in Figure 3H, supplemental figure 6F,J, and we did not observe an impact on behavior in this test. We have edited the text for better clarification:

“..previous studies have implicated *Ucn3* in the PeFA in the regulation of social recognition, stress responsivity, and more recently risk assessment (Venihaki *et al.*, 2004; Deussing *et al.*, 2010; Kuperman *et al.*, 2010; Jamieson *et al.*, 2011; Horii-Hayashi *et al.*, 2021). P. 16

4. Additional clarity about statistical methods and data analyses employed.a. The Methods lack a statistics section. There does not appear to be a description of statistics for behavior data, whether the many comparisons met the assumptions of the many parametric tests, or what software was used for statistics in many of the comparisons. The reviewers do acknowledge that some statistical methods were presented, such as with gene expression data.

Thank you for the very valuable suggestion: we agree that a full-fledged statistics section is necessary to strengthen the manuscript and have now included such section in Methods.

b. Code for Bayesian model of pS6 data could not be located on the indicated GitHub page provided in the manuscript. Either it is not there currently, or it is not well labeled. Please make sure this code is clearly labeled, well annotated, and available prior to publication.

Thanks for pointing out this issue which arose from an incomplete address. We have updated the methods to redirect the to R code: https://gitlab.com/dulaclab/ucn3_neuron_microarray/-/tree/master/braimSourceCode/braim.R

c. In reference to the sex differences uncovered in Figure 1 – Colocalization of c-fos and Ucn3 was significant only in virgin males that attack pups and not females although a trend was observed. Did the authors do a power analysis prior to this study? Is this a power issue or is there a real sex difference here?

Thank you for helping us clarify this issue. We very rarely observed attack in females and included as much data as we were able to collect in the lab, but likely it is underpowered to observe a true sex difference. We have amended the text accordingly on page 6:

“A trend was also observed in infanticidal virgin females, resulting from the analysis of 3 rare infanticidal mice seen out of a large group of otherwise non-aggressive virgin females.”

d. Figure 1 – Supplement 1 Graph E. There are t-test results but please clarify that a t-test was performed in the legend? It appears that a t-test is comparing the colocalization to total Ucn3+ neurons, but if that is the case, an independent t-test is not appropriate.

Thank you for the comment, we have reanalyzed these data with a Chi-Square test and updated the statistics in Figure 1—Supplement 1 panel E.

e. In Figure 7 use caution when using a t-test with only 3 replicates and make sure these data fit the assumptions of this parametric test.

Thank you for the comment, we have double checked that our data satisfy the assumptions for a parametric test.

f. Most of the Figure legends include statistical information but Figure 8 legend is missing n and statistics.

Thank you for pointing out the oversight, we have now updated the figure legends.

g. please ensure each statistic has full reporting, e.g., test statistic, degrees of freedom, in addition to p value.

Thanks for pointing to the need for more statistical details: we have now thoroughly checked and amended all figure legends to ensure full reporting.

5. New images or quantification of images will improve the conclusions in this manuscript.a. Figure 1B is the main figure that demonstrates visualization of enhanced c-Fos reactivity in PeFA of virgin males after infant directed attacks. However, the low power image used does not easily convey this information. While the authors include many higher power images in other subfigures it would be helpful for the reader to see a higher power image inlet in this first subfigure.

We agree that addition of insets in the figure will improve the manuscript, we have now provided high power images in the supplement to figure 1.

b. On page 6 the authors state that excitatory Ucn3 neurons are activated based on the Vglut/Ucn3 colocalization image in Supplemental Figure 1. However, it is difficult to visualize the colocalization in this image. Please provide quantification of the Vglut/Ucn3 positive neurons relative to Ucn3 only positive neurons.

Thank you for suggesting to include the percentage in the main text, we have updated to reflect this percentage on page 5 “91.3%±1.2%”.

c. The images shown in Supplemental Figures 2, 3 and 4 are not ideal for demonstrating that the construct is expressed in the Ucn3 cells. Please show double-staining for the virus-encoded fluorophore and Ucn3 mRNA/protein. Inclusion of quantification for at least one construct would demonstrate the level of expression in Ucn3 neurons.

Thank you for suggesting inclusion of these data. Ucn3 is localized at synapses, so it is not possible to visualize by immunostaining together with the fluorescent tag of the expressed viral construct. Instead, we have now provided *in situ* data for the DREADD construct expression in ucn3 neurons in Supplement to figure 4B. We also have data concerning the impact of chemogenetic stimulation on PeFA^Ucn3^ cell firing in Supplement to Figure 4.

6. Edits to the electrophysiology data in Figure 8 and validation of optogenetic/ chemogenetic protocols.a. Please justify the relevance of including the data in Figure 8A-C as pretty much all brain neurons are expected to receive excitatory input and this data set does not isolate PeFA synapses.b. Using amplitude as a measure of the strength of connectivity appears to be an oversimplification as it is difficult to make a direct comparison between targets using the experimental design in this experiment.

We thank the reviewer for pointing out the lack of clarity in the experimental design underlying this figure. We have included additional information in the figure (Figure 8A) and the figure legends to improve the description of the experiments. We have also included an additional supplemental figures (Figure 3—figure supplement 3) depicting selective targeting of PeFA^Ucn3^ with ChR2 and responses of PeFA^Ucn3^ neurons in response to blue light (465±40 nm) activation. In response to the reviewers’ suggestion, we have replaced Figure 8 A-B in the original submission with raw traces (and average traces Figure 8C) of the evoked events in the neuronal populations of the three projection targets (LS, VMH and AHi) for PeFA^Ucn3^ neurons.

The experiments outlined in Figure 8 were performed to examine the strength of PeFa-Ucn3 projections to three target regions based on the outcome of our tracing experiments. It is true that our experiments were not designed to quantify the precise number of release sites or the precise value of quantal size at PeFA^Ucn3^ target sites. Our data provides a strong evidence of strong connectivity between PeFA Ucn3 neurons and AHi. Specifically, the data in panels C-D show the relative strength of the inputs from PeFA synapses to three different regions. In panel C, we show that we can selectively label and activate PeFA^Ucn3^ projections by expressing ChR2 in Ucn3-Cre mice. By specifically activating PeFA^Ucn3^ projections to LS, VMH and AHi, we were able to ascertain the mean postsynaptic responses in the above-mentioned areas. Our data supports the claim that the mean responses from activating PeFA are significantly stronger in Ahi than those observed in LS or VMH (Figure 8D).

In panel E, we used the paired pulse paradigm to compare the probability of release from PeFA^Ucn3^ projections to LS, VMH and AHi. Previous studies have shown that the paired pulse ratio, the ratio of response to second pulse to that of the first pulse is inversely related to the release probability at many synapses (Manabe et al., 1993; Debanne et al., 1996). Our data strongly supports the claim that the PeFA^Ucn3^ projections to AHi show very high probability of release as evident from significant paired pulse depression (Figure 8E).

c. The authors classify the PSCs they recorded after stimulation of PeFAUcn3 cells as excitatory. This is a fair assumption given that these cells are glutamatergic but the manuscript lacks information specifying that there was pharmacological isolation by gabazine or a similar drug. Please clarify this.

We thank the reviewer for highlighting the lack of clarity in our description. We agree with the reviewer that pharmacological manipulation is one clear way to isolate glutamatergic synaptic connectivity. Another way is to clamp the neurons at the reversal potential for GABAA (Cl^-^). For the inhibitory ionotropic neurotransmitter receptors that carry Cl− (GABAA), under normal slice recording conditions the equilibrium potential is very close to the resting potential of the neurons (approximately –70 mV). Thus, by clamping the neurons at -70 mV during slice recordings, the driving force for the Cl^-^ conduction is 0 (or negligible). Thus, this approach does not necessitate blocking of GABAergic neurotransmission to isolate glutamatergic neurotransmission (EPSC) (Kato et al., 2011).

d. Figure 8A-C: As noted above there is uncertainty as to the value of these data in the context of the current study. However, if they are included then please also include raw traces and not only quantification.

We have replaced the original 8A-B with the raw traces (and average traces) of the evoked events in the neuronal populations of the three projection targets (LS, VMH and AHi) for PeFA^Ucn3^ neurons. We have also included additional schematic in the figure (Figure 8A) and the figure legends to improve the description of the experiments. For details, please see the answer to comments 6a and 6b.

e. Please include validation (eg by slice electrophysiology and application of light/CNO) that show that the opto- and chemogenetic strategies for stimulation and inhibition achieves the in vivo stimulation goals at a cellular level.

We have updated the manuscript and have included additional supplementary figures (Figure 3—figure supplement 3E-G and Figure 4—figure supplement 4C-E) to show optogenetic and chemogenetic stimulation of PeFA^Ucn3^ neurons by light/CNO.

References

Cox, D.W. and Bachelard, H.S. (1982) Attenuation of evoked field potentials from dentate granule cells by low glucose, pyruvate + malate, and sodium fluoride. Brain research, 239, 527-534.

Debanne, D., Guerineau, N.C., Gahwiler, B.H. and Thompson, S.M. (1996) Paired-pulse facilitation and depression at unitary synapses in rat hippocampus: quantal fluctuation affects subsequent release. J Physiol, 491 ( Pt 1), 163-176.

Deussing, J.M., Breu, J., Kuhne, C., Kallnik, M., Bunck, M., Glasl, L., Yen, Y.C., Schmidt, M.V., Zurmuhlen, R., Vogl, A.M., Gailus-Durner, V., Fuchs, H., Holter, S.M., Wotjak, C.T., Landgraf, R., de Angelis, M.H., Holsboer, F. and Wurst, W. (2010) Urocortin 3 modulates social discrimination abilities via corticotropin-releasing hormone receptor type 2. The Journal of neuroscience : the official journal of the Society for Neuroscience, 30, 9103-9116.

Dong, W.Q., Schurr, A., Reid, K.H., Shields, C.B. and West, C.A. (1988) The rat hippocampal slice preparation as an in vitro model of ischemia. Stroke, 19, 498-502.

Jamieson, P.M., Cleasby, M.E., Kuperman, Y., Morton, N.M., Kelly, P.A., Brownstein, D.G., Mustard, K.J., Vaughan, J.M., Carter, R.N., Hahn, C.N., Hardie, D.G., Seckl, J.R., Chen, A. and Vale, W.W. (2011) Urocortin 3 transgenic mice exhibit a metabolically favourable phenotype resisting obesity and hyperglycaemia on a high-fat diet. Diabetologia, 54, 2392-2403.

Kato, G., Kosugi, M., Mizuno, M. and Strassman, A.M. (2011) Separate inhibitory and excitatory components underlying receptive field organization in superficial medullary dorsal horn neurons. The Journal of neuroscience : the official journal of the Society for Neuroscience, 31, 17300-17305.

Kohl, J., Babayan, B.M., Rubinstein, N.D., Autry, A.E., Marin-Rodriguez, B., Kapoor, V., Miyamishi, K., Zweifel, L.S., Luo, L., Uchida, N. and Dulac, C. (2018) Functional circuit architecture underlying parental behaviour. Nature, 556, 326-331.

Kuperman, Y., Issler, O., Regev, L., Musseri, I., Navon, I., Neufeld-Cohen, A., Gil, S. and Chen, A. (2010) Perifornical Urocortin-3 mediates the link between stress-induced anxiety and energy homeostasis. Proc Natl Acad Sci U S A, 107, 8393-8398.

Manabe, T., Wyllie, D.J., Perkel, D.J. and Nicoll, R.A. (1993) Modulation of synaptic transmission and long-term potentiation: effects on paired pulse facilitation and EPSC variance in the CA1 region of the hippocampus. J Neurophysiol, 70, 1451-1459.

Margrie, T.W., Sakmann, B. and Urban, N.N. (2001) Action potential propagation in mitral cell lateral dendrites is decremental and controls recurrent and lateral inhibition in the mammalian olfactory bulb. Proc Natl Acad Sci U S A, 98, 319-324.

McNay, E.C. and Gold, P.E. (1999) Extracellular glucose concentrations in the rat hippocampus measured by zero-net-flux: effects of microdialysis flow rate, strain, and age. J Neurochem, 72, 785-790.

Schurr, A., West, C.A. and Rigor, B.M. (1989) Electrophysiology of energy metabolism and neuronal function in the hippocampal slice preparation. J Neurosci Methods, 28, 7-13.

Spruston, N., Jonas, P. and Sakmann, B. (1995) Dendritic glutamate receptor channels in rat hippocampal CA3 and CA1 pyramidal neurons. J Physiol, 482 ( Pt 2), 325-352.

Venihaki, M., Sakihara, S., Subramanian, S., Dikkes, P., Weninger, S.C., Liapakis, G., Graf, T. and Majzoub, J.A. (2004) Urocortin III, a brain neuropeptide of the corticotropin-releasing hormone family: modulation by stress and attenuation of some anxiety-like behaviours. J Neuroendocrinol, 16, 411-422.

Wittmann, G., Fuzesi, T., Liposits, Z., Lechan, R.M. and Fekete, C. (2009) Distribution and axonal projections of neurons coexpressing thyrotropin-releasing hormone and urocortin 3 in the rat brain. J Comp Neurol, 517, 825-840.

Wu, Z., Autry, A.E., Bergan, J.F., Watabe-Uchida, M. and Dulac, C.G. (2014) Galanin neurons in the medial preoptic area govern parental behaviour. Nature, 509, 325-330.